# The CDT of *Helicobacter hepaticus* induces pro-survival autophagy and nucleoplasmic reticulum formation concentrating the RNA binding proteins UNR/CSDE1 and P62/SQSTM1

**Wencan He**[1☯]**, Lamia Azzi-Martin**[1,2☯]**, Valérie Velasco**[3]**, Philippe Lehours**[1,4]**,
Pierre Dubus**[1,2,5]**, Mojgan Djavaheri-Mergny**[6,7]**, Armelle Ménard**[1] *

**1** Univ. Bordeaux, INSERM, UMR1053 Bordeaux Research in Translational Oncology, BaRITOn, Bordeaux, France, **2** Univ. Bordeaux, UFR des Sciences Médicales, Bordeaux, France, **3** Département de BioPathologie, Service d'Anatomie Pathologique, Plateforme de Pathologie Expérimentale, Institut Bergonié, Bordeaux, France, **4** CHU de Bordeaux, Laboratoire de Bactériologie, Centre National de Référence des Campylobacters et Hélicobacters, Bordeaux, France, **5** CHU de Bordeaux, Institut de Pathologie et de Biologie du Cancer, Bordeaux, France, **6** Centre de Recherche des Cordeliers (CRC), Université de Paris, Sorbonne Université, INSERM, Institut Universitaire de France, Paris, France, **7** Metabolomics and Cell Biology Platforms, Institut Gustave Roussy, Villejuif, France

☯ These authors contributed equally to this work.
* armelle.menard@u-bordeaux.fr

## Abstract

Humans are frequently exposed to bacterial genotoxins of the gut microbiota, such as colibactin and cytolethal distending toxin (CDT). In the present study, whole genome microarray-based identification of differentially expressed genes was performed *in vitro* on HT29 intestinal cells while following the ectopic expression of the active CdtB subunit of *Helicobacter hepaticus* CDT. Microarray data showed a CdtB-dependent upregulation of transcripts involved in positive regulation of autophagy concomitant with the downregulation of transcripts involved in negative regulation of autophagy. CdtB promotes the activation of autophagy in intestinal and hepatic cell lines. Experiments with cells lacking autophagy related genes, ATG5 and ATG7 infected with CDT- and colibactin-producing bacteria revealed that autophagy protects cells against the genotoxin-induced apoptotic cell death. Autophagy induction could also be associated with nucleoplasmic reticulum (NR) formation following DNA damage induced by these bacterial genotoxins. In addition, both genotoxins promote the accumulation of the autophagic receptor P62/SQSTM1 aggregates, which colocalized with foci concentrating the RNA binding protein UNR/CSDE1. Some of these aggregates were deeply invaginated in NR in distended nuclei together or in the vicinity of UNR-rich foci. Interestingly, micronuclei-like structures and some vesicles containing chromatin and γH2AX foci were found surrounded with P62/SQSTM1 and/or the autophagosome marker LC3. This study suggests that autophagy and P62/SQSTM1 regulate the abundance of micronuclei-like structures and are involved in cell survival following the DNA damage induced by CDT and colibactin. Similar effects were observed in response to DNA

**Data Availability Statement:** All relevant data are within the manuscript and its Supporting Information files.

**Funding:** WH was the recipient of a pre-doctoral fellowship from the China Scholarship Council Scholarships. This work was supported in part by the University of Bordeaux and INSERM. The funders had no role in study design, data collection and analysis, decision to publish, or preparation of the manuscript.

**Competing interests:** The authors have declared that no competing interests exist.

damaging chemotherapeutic agents, offering new insights into the context of resistance of cancer cells to therapies inducing DNA damage.

## Author summary

The mucosal epithelium is a common target of damage induced by chronic bacterial infections and their toxins. Cytolethal Distending Toxin-secreting bacteria and colibactin-producing bacteria are frequently found in the human digestive microbiota and their toxins trigger potent DNA damage. Here, we showed the activation of autophagy following the genotoxic stress induced by these toxins. Autophagy led to selective removal of genotoxin-induced micronuclei-like structures and protected the cells against the genotoxin-induced apoptotic cell death. Under these conditions, the induction of autophagy could also be associated with the formation of transient messenger-rich ribonucleoprotein particles concentrating the autophagic receptor P62/SQSTM1 clustered in the genotoxin-induced nucleoplasmic reticulum. P62/SQSTM1 plays a central role in this pro-survival autophagy, which deserves to be investigated. Similar responses were observed with some DNA damaging agents used during chemotherapies, suggesting that autophagy together with nucleoplasmic reticulum formation following DNA damage may contribute to the resistance of cancer cells to therapies inducing DNA damage.

## Introduction

Bacterial genotoxins, cytolethal distending toxin (CDT) and colibactin are frequently identified in bacteria associated with digestive pathologies. CDT is an $AB_2$ toxin composed of 3 subunits (A: CdtB; $B_2$: CdtA; and CdtC) of which the binding moiety comprised of CdtA and CdtC allows the internalization of CdtB which is the active subunit and the most conserved of the subunits among CDT-secreting bacteria [1]. Colibactin is a complex secondary metabolite produced mainly by some genotoxic *Escherichia coli* strains from the phylogenetic group B2 containing a polyketide synthase machinery (pks genomic island, pks+ *E. coli*) [1]. Both genotoxins induce DNA damage, activating the checkpoint responses and leading to cell cycle arrest to allow DNA repair and cell survival [2]. If improperly repaired, DNA damage leads to genomic instability and deregulation of cellular functions, potentially leading to cancer. If the damage is beyond repair, cells undergo either apoptosis or senescence.

Upon DNA damage, autophagy and the DNA damage response are activated. Both processes are essential for cellular homeostasis and survival. In this context, CDT-induced DNA damage promotes autophagy *in vitro*, as evidenced by an increased level of autophagosome marker LC3 [3,4], the accumulation of autophagosomes and the stimulation of the autophagic flux [4]. All of these effects are absent using a CDT mutant strain unable to trigger DNA double-strand breaks (DSBs), suggesting the requirement of DSBs. In line with that, it was recently shown that colonization of non-tumoral colonic mucosa from patients with colorectal cancer with colibactin-producing *E. coli* is associated with high autophagy-related messenger RNA (mRNA) levels [5]. *In vitro*, colibactin also promotes autophagy in human colon cancer cells that is necessary for DNA damage repair (DDR) [5]. In the ApcMin/+mice model, colibactin-induced autophagy is required to prevent colorectal tumorigenesis [5]. Recent evidence supports the idea that autophagy induced by colibactin is not only involved in bacterial

degradation but also in genotoxic damage repair [5]. However, little work has focused on the role of autophagy in regulating cell death induced during genotoxin-secreting bacterial infection.

The nuclear remodeling resulting from the DNA damage induced by CDT and colibactin can promote the formation of nucleoplasmic reticulum (NR), deeply invaginated in the nucleoplasm of giant nuclei in surviving cells [6]. CDT-induced NR formation was observed both *in vivo* and *in vitro*. The core of these NR concentrate protein production machinery of the cell, as well as controlling elements of protein turnover. Indeed, NR are active sites of mRNA translation and they concentrate ribosomes, polyadenylated RNA, proteins involved in mRNA translation (eIF4F complex), and the main components of the complex mCRD involved in mRNA turnover [6]. The CDT-intoxicated giant polyploid cells accumulate NR, survive and keep proliferating with an apparent NR resorption and return to normal size a few days after intoxication, suggesting that insulation and concentration of these transient and reversible adaptive ribonucleoprotein particles within the nucleus are highly dynamic and allow the cell to pause and repair the DNA damage caused by bacterial genotoxins in order to maintain cell survival. In this context, pro-survival autophagy may occur in response to CDT and may play a role in genotoxin-induced NR formation in surviving cells.

The present study aimed to define the role of autophagy in regulating the cell death/survival balance in the context of CDT intoxication. Microarray-based identification of differentially expressed genes involved in autophagy was performed following the lentiviral expression system of the CdtB subunit of *H. hepaticus* in intestinal epithelial cells [7]. Then, autophagy markers were evaluated using a 2-way original system previously validated [7] comprised of (1) coculture experiments with an *H. hepaticus* strain and its corresponding ΔCDT isogenic mutant strain to examine non-CDT bacterial factors in the effects observed and (2) direct cellular ectopic expression of *H. hepaticus* CdtB and its corresponding mutated CdtB (H265L) lacking catalytic activity to examine the effects specifically related to the CdtB. The effects of CDT/CdtB on autophagy were assessed on human intestinal and hepatic epithelial cell lines, as *H. hepaticus* colonizes primarily the intestine and the liver. CDT-induced NR formation in the context of autophagy was also investigated. Cells were treated with two pharmacological inhibitors of autophagy (bafilomycin A1 and chloroquine); ATG5 and ATG7 silencing was performed using the CRISPR-Cas9 System; then the consequences of autophagy inhibition were evaluated on bacterial genotoxin effects.

## Results

### Autophagy-associated genes are regulated in response to the CdtB of *Helicobacter hepaticus*

The global expression of human genes was quantified in transduced epithelial intestinal HT29 cells using whole genome microarrays following ectopic expression of the active CdtB subunit of the CDT of *H. hepaticus versus* the control tdTomato fluorescent protein (TFP) [7]. This lentiviral approach was previously validated [8] since the well-known cytopathogenic effects associated with the CdtB were observed, *i.e.* actin cytoskeleton remodeling, cellular and nuclear distension, vinculin delocalization and formation of cortical actin-rich large lamellipodia [8–10]. Among autophagy-related genes from the Human Autophagy Database (raw data presented in S1 Table, Sheet 1) and included in those microarrays (Fig 1 and S2 Table), numerous genes involved in positive regulation of autophagy were significantly upregulated upon CdtB expression, except for ATG16L2, PARK2, STK11 and ULK2 genes whose expression was decreased. CdtB-upregulated transcripts encode proteins involved in different events during the formation of the autophagosomes [11], such as ULK1, ATG13 and RB1CC1/FIP200

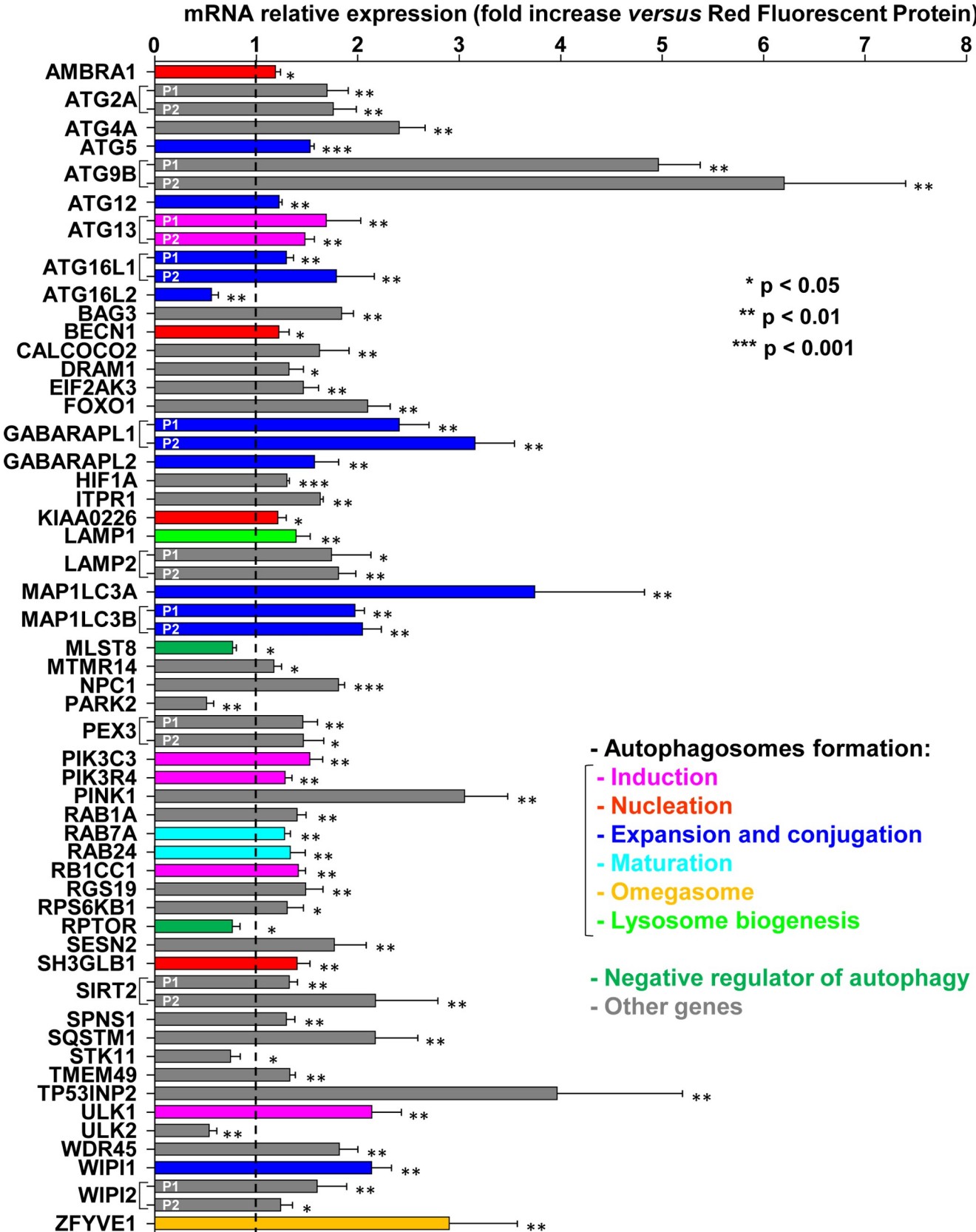

**Fig 1. Microarray-based identification of differentially expressed autophagy-related genes in response to *Helicobacter hepaticus* CdtB in intestinal epithelial cells.** The expression of genes was determined in HT29 intestinal cells using the Human GE 4x44K v2 Microarray Kit (Agilent Technologies) after a 72 h transduction with lentiviral particles expressing the CdtB of *H. hepaticus* strain 3B1 *versus* the tdTomato fluorescent protein (TFP) as previously described [7]. The relative expression of genes in response to CdtB is reported as a fold change *versus* the value for cells cultured with lentiviral particles expressing the TFP. Results are presented as the mean of 4 replicates as 4 independent transduction experiments were performed. The data presented for ATG5, HIF1A and NPC1 are the results of 40 replicates as 10 probes for each mRNA of these genes were included on the Microarray Kit. The list of genes related in autophagy to be checked in the Microarray data was first extracted from the Human Autophagy Database (HADb, http://autophagy.lu/clustering/ ). Then a subsequent selection based on their autophagic status annotation available in The Human Gene Database was applied in order to select the major genes involved in autophagy. The discontinuous line shows the basal rate in cells expressing TFP. Asterisks denote significant results. P1 and P2 represent the 2 probe names (S2 Table) used for mRNA quantification. Details are presented in S2 Table (name and sequence of the probes, the corresponding gene name, the genbank accession number, the locus and the transcript variant). Abbreviations: AMBRA1, Autophagy and Beclin-1 Regulator 1; ATG, Autophagy Related Gene; ATG16L, Autophagy Related 16 Like Gene; BAG3, BAG Cochaperone 3;BECN1/Vps30/ATG6, Beclin 1; CALCOCO2, Calcium Binding And Coiled-Coil Domain 2; DRAM1, DNA-damage Regulated Autophagy Modulator 1; eiF2AK3, Eukaryotic Translation Initiation Factor 2-Alpha Kinase 3; FOXO1, Forkhead Box O1; GABARAPL, GABA(A) Receptor-Associated Protein Like; HIF1A, Hypoxia Inducible Factor 1 Subunit Alpha; ITPR1, Inositol 1,4,5-Triphosphate Receptor, type 1; KIAA0226/RUBCN, Rubicon Autophagy Regulator; LAMP1, Lysosomal Associated Membrane Protein 1; LAMP2, Lysosomal Associated Membrane Protein 2; MAP1LC3A, Microtubule Associated Protein 1 Light Chain 3 Alpha; MAP1LC3B, Microtubule Associated Protein 1 Light Chain 3 Beta; MLST8, MTOR (Mechanistic Target of Rapamycin Kinase) Associated Protein, LST8 Homolog (mammalian lethal with Sec13 protein); MTMR14, MyoTubularin Related Protein 14; NPC1, NPC Intracellular Cholesterol Transporter 1 (Niemann-Pick disease, type C1); PARK2/PARKN, Parkin RBR E3 Ubiquitin Protein Ligase; PEX3, Peroxisomal Biogenesis Factor 3; PIK3C3//Vps34, Phosphoinositide-3-Kinase, class 3; PIK3R4/Vps15, Phosphoinositide-3-Kinase, Regulatory subunit 4; PINK1; Phosphatase and Tensin Homolog (PTEN) Induced Kinase 1; RAB1A, Member RAS Oncogene Family; RAB7A, Member RAS Oncogene Family; RAB24, member RAS oncogene family; RB1CC1/FIP200, RB1 (RetinoBlastoma Transcriptional Corepressor 1) Inducible Coiled-Coil 1; RGS19, Regulator of G-protein Signaling 19; RPS6KB1, Ribosomal Protein S6 Kinase B1; RPTOR, Regulatory Associated Protein Of MTOR Complex 1; SESN2, Sestrin 2; SH3GLB1/BIF1, SH3 Domain Containing Growth Factor Receptor Bound Protein 2 (GRB2) Like, Endophilin B1; SIRT2, Sirtuin 2; SPNS1, Sphingolipid Transporter 1 (Putative); P62/SQSTM1, Sequestosome 1; STK11; Serine/Threonine Kinase 11; TMEM49/VMP1, Vacuole Membrane Protein 1; TP53INP2, Tumor Protein P53 Inducible Nuclear Protein 2; ULK, unc-51-like kinase; WD, tryptophan-aspartic acid; WDR45/WIPI4, WD (tryptophan-aspartic acid) Repeat Domain 45; WIPI, WD Repeat Domain, Phosphoinositide Interacting; ZFYVE1/DFCP1, Zinc Finger FYVE-Type Containing 1.

(induction); PIK3C3/Vps34 and PIK3R4/Vps15 (class III phosphatidylinositol 3-kinase complex I); BECN1/Vps30/ATG6, SH3GBL1/BIF1, KIAA0226/RUBCN and AMBRA1 (nucleation), ATG5, ATG12, ATG16L1, GABARAPL1/2, MAP1LC3A/B, WIPI1, (expansion and conjugation); RAB7A and RAB24 (maturation); and P62/SQSTM1 (cargo degradation/recycling). Other genes related to autophagy/lysosome processes were also regulated by CdtB, such as ZFYVE1/DFCP1 (omegasome, the site where phagophores form) or LAMP1 (lysosome biogenesis). As expected, RPTOR (Regulatory associated Protein of mTOR complex 1) and MLST8 (target of the rapamycin complex subunit LST8) mRNA levels were downregulated, as their respective encoded proteins are involved in negative regulation of autophagy.

Autophagy and apoptosis are intimately interconnected and many ATGs are recognized and cleaved by caspases [12]. It is thus not surprising that the CdtB of *H. hepaticus* also regulated certain genes involved in the 'apoptosis and autophagy' pathway (S1A Fig and S3 Table). As expected, CdtB intoxication led to an increase in the mRNA level of apoptosis-regulator proteins, BID and caspase-3, as well as those from inflammatory caspase-1 and -4 involved in pyroptosis during host-pathogen interaction [13]. CdtB upregulation of Caspase 1 protein was also confirmed using xenograft mouse models (S1B Fig). Raw data of inflammation-related genes (TFP and CdtB) are presented in S1 Table (Sheet 2).

CdtB also upregulates some transcripts encoding proteins involved in the positive regulation of autophagy (S1A Fig and S3 Table), such as those of BIRC5/Survivin, the autophagy-induced DNA damage suppressor [14]; cathepsin D (CTSD) that can function as an anti-apoptotic mediator by inducing autophagy under cellular stress [15]; cathepsin L1 (CTSL1), a key member of the lysosomal protease family that facilitates autophagy and proteasomal protein processing [16]; and PEA-15 (proliferation and apoptosis adaptor protein 15), an inducer of autophagy associated with cell survival [17].

Taken together, the upregulation of the expression of numerous genes involved in positive regulation of autophagy concomitant with the downregulation of RPTOR and MLST8 mRNA

                                                                                      

involved in negative regulation of autophagy suggest an activation of the autophagy pathway in response to CdtB intoxication.

## CDT, *via* its active CdtB subunit, induces autophagy

Upon autophagy, the microtubule-associated protein 1A/1B light chain 3 (LC3) is conjugated to phosphatidylethanolamine to yield LC3-II on the surface of nascent autophagosomes. Autophagosome numbers are widely scored by evaluation of either LC3-II puncta or LC3-II expression levels. Autophagosome numbers were thus assessed by quantifying endogenous LC3 puncta numbers through immunofluorescence microscopy with a specific LC3 antibody during coculture experiments with *H. hepaticus* strain and its corresponding CDT-knockout (ΔCDT) mutant strain (Fig 2A). Compared to non-infected cells, an increase in LC3 puncta was measured in Hep3B cells infected with *H. hepaticus*, while the increase in LC3 puncta was almost absent in cells infected with the ΔCDT mutant strain (Fig 2A), suggesting that the CDT is most likely the main virulence factor associated with LC3 puncta increase. No CDT effects were observed upon *H. hepaticus* infection of HT29 cells. This cell line had already been reported to be resistant to CDT effects after *Helicobacter pullorum* infection but became susceptible to CdtB by using direct expression of the toxin in the cell [8].

Similarly, LC3 puncta were thus quantified in epithelial intestinal HT29 and hepatic Hep3B cells expressing the red fluorescent protein (RFP), the CdtB subunit of the CDT of *H. hepaticus* (CdtB) or the CdtB of *H. hepaticus* with the H265L mutation lacking catalytic activity. In both cell lines (Figs 2B1 and 2C1 and S2A), the level of LC3 puncta was strongly increased upon expression of *H. hepaticus* CdtB *versus* control RFP. No significant increase in LC3 puncta was observed in response to the mutant form of *H. hepaticus* CdtB harboring a His→Leu mutation at residue 265 (H265L) that is crucial for CdtB catalytic activity [7], indicating that LC3 puncta increase is attributed to the CdtB. Time-course western blot analysis of the protein level of LC3 also confirmed the accumulation of LC3-II upon expression of *H. hepaticus* CdtB (Fig 3A).

In order to understand the role of autophagy in response to CdtB intoxication, cells were treated with two well-known pharmacological inhibitors of autophagy flux, bafilomycin A1 and chloroquine, and LC3 puncta were subsequently quantified. Both compounds significantly increased LC3 puncta upon CdtB expression in HT29 and Hep3B cell lines (Figs 2B2 and 2C2 and S2B) again confirming the activation of autophagy flux upon CdtB expression.

Autophagic flux is defined as a measure of the amount of cellular material degraded by the autophagy process. The exogeneous fluorescent tandem tagged mCherry-GFP-LC3 monomeric protein was used to monitor autophagic flux with subsequent determination of the ratio between the number of red (mCherry$^+$/GFP$^-$, autophagolysosomes) and yellow (mCherry$^+$/GFP$^+$, autophagosomes) dots [18]. The basis of this method lies in higher sensitivity of the GFP signal to the acidic pH environment of the lysosome compared to the mCherry signal [18]. As transgenic cells expressing the RFP could not be evaluated using this assay, transgenic cells expressing the CdtB *versus* CdtB-H265L were used. In both HT29 and Hep3B cell lines, CdtB strongly stimulated the appearance of red LC3 puncta, compared to CdtB-H265L, indicating activation of the whole autophagy flux in response to CdtB (Figs 2B3 and 2C3 and S2C).

Taken together, these data showed that CDT, *via* its active CdtB subunit, promotes autophagy activation in cells. CdtB also activates AMP-activated protein kinase (AMPK), a key sensor of autophagy as evidenced by an increase in the phosphorylation of AMPK (on Thr172, Fig 3B).

## CdtB upregulates the expression of P62/SQSTM1

P62/SQSTM1 is an autophagy substrate that is involved in the selective transport of cargo to the autophagosomes. It is also used as a reporter of selective autophagy activity. The level of

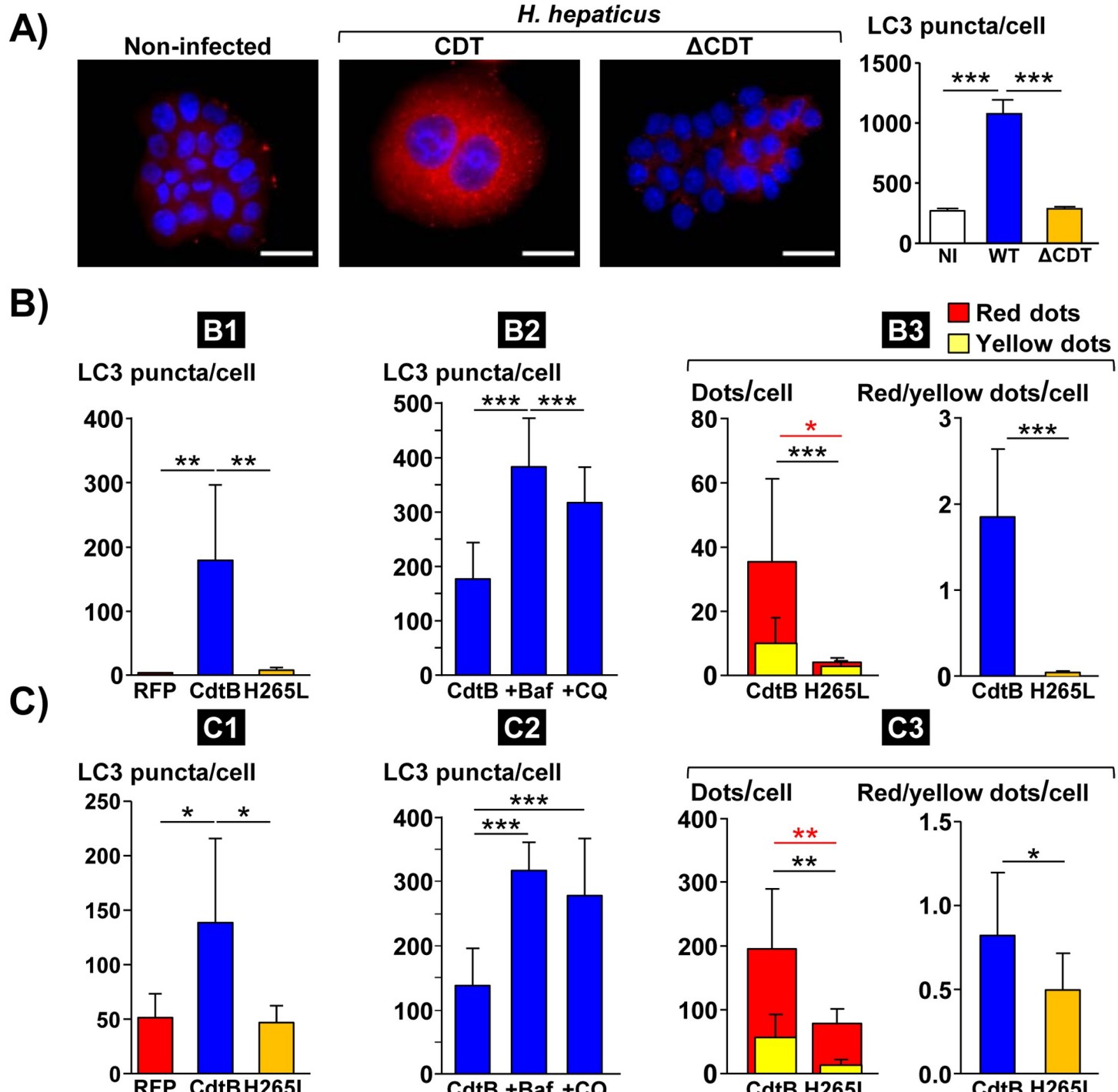

**Fig 2. Effects of *Helicobacter hepaticus* cytolethal distending toxin on LC3 expression in human intestinal and hepatic epithelial cells. A)** Analysis of LC3 puncta in hepatic Hep3B cells infected for 3 days with *H. hepaticus* and its corresponding ΔCDT mutant strain. These cells were processed for fluorescent labeling of LC3 (red) and a counterstaining with DAPI (blue). Fluorescent staining was observed using wide field fluorescence imaging. Transgenic HT29 **(B)** and Hep3B **(C)** cells were cultivated with doxycycline for 72 h to induce the expression of the control Red Fluorescent Protein (RFP), the CdtB of *H. hepaticus* strain 3B1 or the CdtB of *H. hepaticus* strain 3B1 with the H265L mutation which has no catalytic activity [21]. These cell lines were also treated with bafilomycin A1 (30 nM) or chloroquine (30 μM) 48 h and 24 h after doxycycline induction for a duration of 24 h and 48 h, respectively. Then, cells were processed for fluorescent staining with primary antibodies generated against LC3 associated with fluorescent labeled-secondary antibodies (green) and DAPI to counterstain the nuclei (blue) (S2A, S2B Fig). Autophagic flux was also measured in those cells expressing the tandem-tagged mCherry-GFP-LC3 protein with subsequent yellow (mCherry+/GFP+) and red (mCherry+/GFP-) dot/puncta counting (yellow dots) (S2C Fig) [18]. The number of fluorescent LC3 puncta was quantified using the "Find Maxima" function of ImageJ. The results are presented as the mean in one representative experiment (performed in triplicate) out of three. A minimum of 500 cells were measured. **B)** Analysis of LC3 puncta in transgenic HT29 cells. **B1)** RFP-,

CdtB- and H265L-expressing cells. **B2)** CdtB-expressing cells treated with bafilomycin A1 or chloroquine. **B3)** autophagic flux measured in CdtB- and H265L-expressing cells. **C)** Analysis of LC3 puncta in transgenic Hep3B cells. **C1)** RFP-, CdtB- and H265L-expressing cells. **C2)** CdtB-expressing cells treated with bafilomycin A1 or chloroquine. **C3)** autophagic flux measured in CdtB- and H265L-expressing cells. *p<0.05, **p< 0.01, ***p< 0.001. Abbreviations: Baf., bafilomycin A1; CdtB, CdtB of *H. hepaticus* strain 3B1; CQ, chloroquine; DAPI, 4', 6'-diamidino-2-phenylindol; ΔCDT, CDT isogenic mutant of *H. hepaticus* strain 3B1; H265L, *H. hepaticus* CdtB with the mutation His→Leu at residue 265 involved in catalytic activity; NI, non-infected; RFP, Red fluorescent protein; WT, *H. hepaticus* strain 3B1 = wild type strain.

P62/SQSTM1 mRNA was upregulated upon CdtB intoxication (Fig 1). CdtB also led to an increase in the expression of P62/SQSTM1 protein, as well as its phosphorylated (Ser403) form (Fig 3A and 3B). The increase in P62/SQSTM1 upon CdtB intoxication was also confirmed by

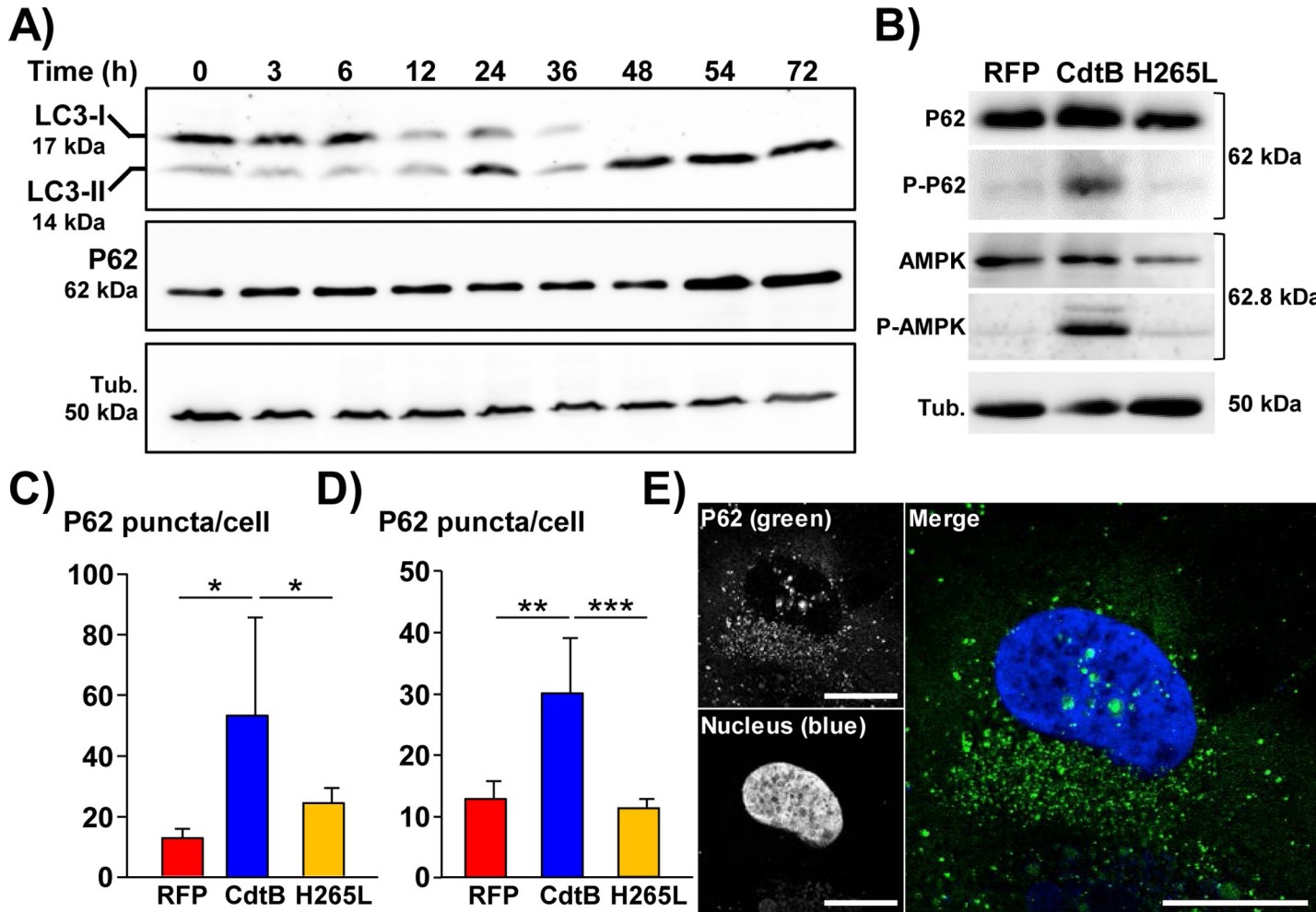

**Fig 3. Effects of *Helicobacter hepaticus* cytolethal distending toxin on LC3, P62/SQSTM1 and AMPK expression in human epithelial cells.** Transgenic HT29 and Hep3B cells were cultivated with doxycycline for 72 h to induce the expression of the control Red Fluorescent Protein (RFP), the CdtB of *H. hepaticus* strain 3B1 or the CdtB of *H. hepaticus* strain 3B1 with the H265L mutation which has no catalytic activity [21]. Then, cells were processed for western blot analysis or fluorescent staining with primary antibodies generated against P62/SQSTM1 associated with fluorescent labeled-secondary antibodies (green) and DAPI to counterstain the nuclei (blue) (S2A Fig). The number of fluorescent P62/SQSTM1 bodies was quantified using the "Find Maxima" function of ImageJ. The results are presented as the mean in one representative experiment (performed in triplicate) out of three. A minimum of 500 cells were measured. **A)** Time-course western blot analysis of the protein expression level of LC3 and P62/SQSTM1 in response to the CdtB of *H. hepaticus* in HT29 cells. **B)** The protein expression level and phosphorylation status of P62/SQSTM1 and AMPK in response to RFP, CdtB and H265L were analyzed by western blot in transgenic HT29 cells. **(C)** Quantification of P62/SQSTM1 bodies in transgenic HT29 cells. **(D)** Quantification of P62/SQSTM1 bodies in transgenic Hep3B cells. **E)** Confocal image of Hep3B transgenic cells expressing the CdtB of *H. hepaticus* strain 3B1 (72 h) processed for P62/SQSTM1 fluorescent staining (green) and DAPI (blue). Scale bar, 10 μm. *p<0.05, **p< 0.01, ***p< 0.001. Abbreviations: AMPK, AMP-activated protein kinase; CdtB, CdtB of *H. hepaticus* strain 3B1; DAPI, 4', 6'-diamidino-2-phenylindol; H265L, *H. hepaticus* CdtB with the mutation His→Leu at residue 265 involved in catalytic activity; P-AMPK, phosphorylated AMP-activated protein kinase; P62, P62/SQSTM1; P-P62, phosphorylated P62/SQSTM1; RFP, Red fluorescent protein; Tub., tubulin.

immunofluorescence microscopy with subsequent quantification (Fig 3C and 3D). In addition, confocal analysis revealed P62/SQSTM1 bodies in the giant nuclei of Hep3B cells upon CdtB expression (Fig 3E), supporting the idea that P62/SQSTM1 may shuttle to the nucleus in response to CdtB intoxication. Indeed, P62/SQSTM1 can shuttle between the nucleus and cytoplasm to bind with ubiquitinated cargos and facilitate nuclear and cytosolic protein quality control [19,20]. To exclude the possibility that this observation could be a side effect of 2-dimensionnal cell culture, *in vivo* models were used to investigate the presence of nuclear P62/SQSTM1. Xenograft mouse models were used [21]. HT29 and Hep3B CdtB-derived tumors were compared to control RFP-derived tumors. LC3 staining was first performed and quantification revealed the increase in LC3 in engrafted HT29- and Hep3B-CdtB-derived mice, compared to RFP-derived mice (Fig 4A). P62/SQSTM1 staining revealed an increase in

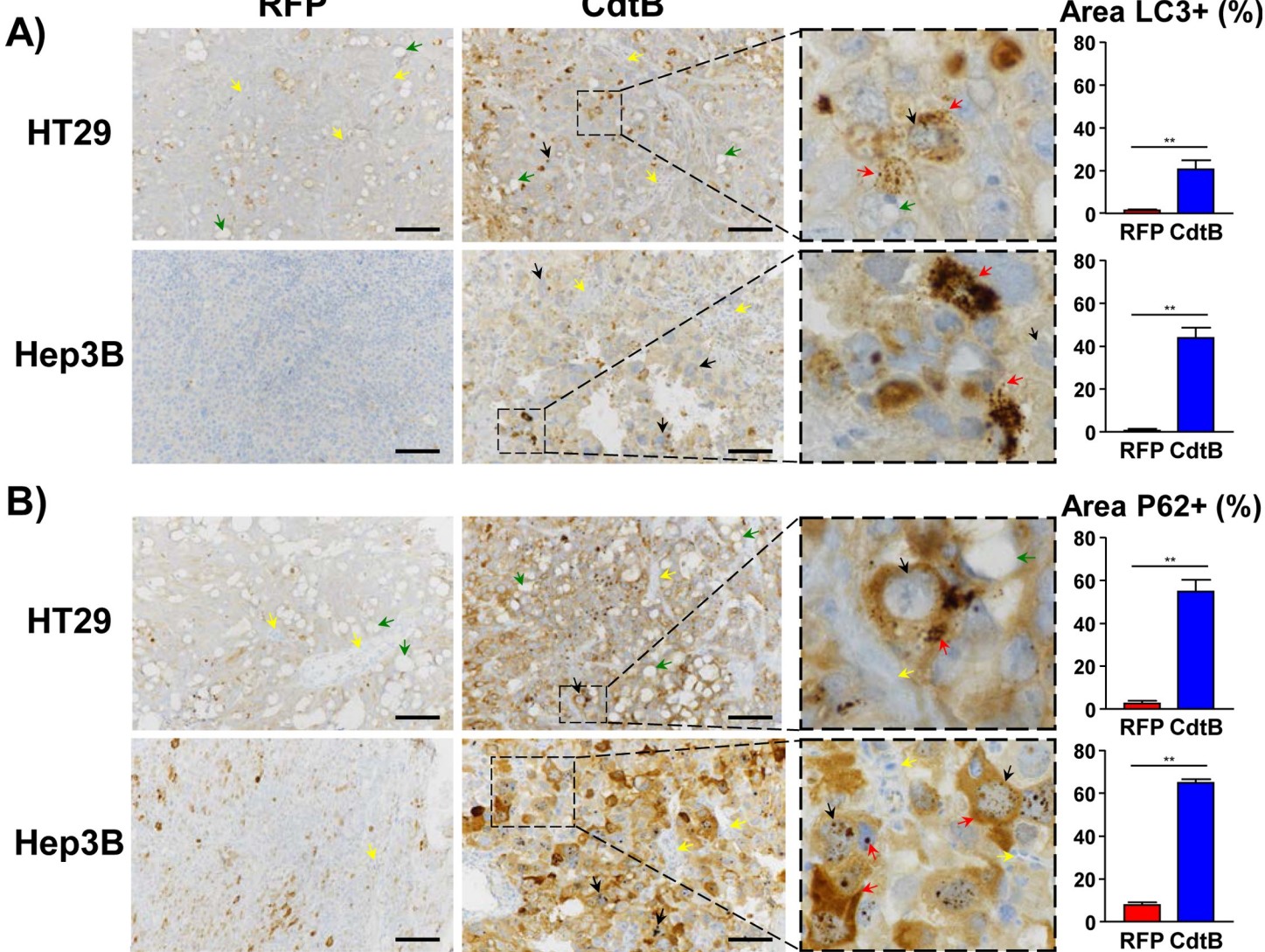

**Fig 4. Effect of *Helicobacter hepaticus* CdtB expression on LC3 and P62 expression in engrafted cells.** HT29- and Hep3B-transgenic cell lines were engrafted into immunodeficient mice as previously reported [21]. Three μm-tissue sections of the xenograft-derived tumors were prepared from formalin-fixed paraffin-embedded tissues and submitted to standard hematoxylin staining and immunostaining raised against LC3 (**A**) and P62/SQSTM1 (**B**). Boxes correspond to enlargement. The yellow, black, green and red arrows represent the murine infiltrates, the giant cells, the mucins (HT29) and the LC3 puncta or P62/SQSTM1 bodies, respectively. Quantification was performed by using the 'Threshold' function of ImageJ (v. 1.52n). Scale bar, 50 μm. **p< 0.01. Scale bar, 50 μm. Abbreviations: CdtB of *H. hepaticus* strain 3B1; P62, P62/SQSTM1; RFP, Red fluorescent protein.

diffuse cytoplasmic P62/SQSTM1 staining, as well as the presence of P62/SQSTM1 bodies in the cytoplasm (Fig 4B) in both HT29- and Hep3B-CdtB-derived tumor cells. Furthermore, the labeling of P62/SQSTM1 from CdtB-Hep3B-derived tumor suggests the presence of nuclear P62/SQSTM1 bodies (magnified area in the box of Fig 4B).

### Cytoplasmic P62/SQSTM1 aggregates are invaginated in distended nuclei in response to CdtB

To further confirm the presence of nuclear P62/SQSTM1, nuclear lamina and P62/SQSTM1 immunofluorescent co-staining was performed upon CdtB intoxication in Hep3B cells. As shown in Fig 5, CdtB promotes the accumulation of P62/SQSTM1 bodies in the cytosol. In addition, some P62/SQSTM1 bodies were found outside of the nuclei and seemed to be engulfed in the nuclear lamina, suggesting that P62/SQSTM1 bodies are not nuclear but invaginated in nucleosome of CdtB-giant cells (Fig 5).

It was shown that CDT induced-nuclear remodeling can be associated with the formation of deep cytoplasmic invaginations in the nuclei of giant cells. These invaginations also known

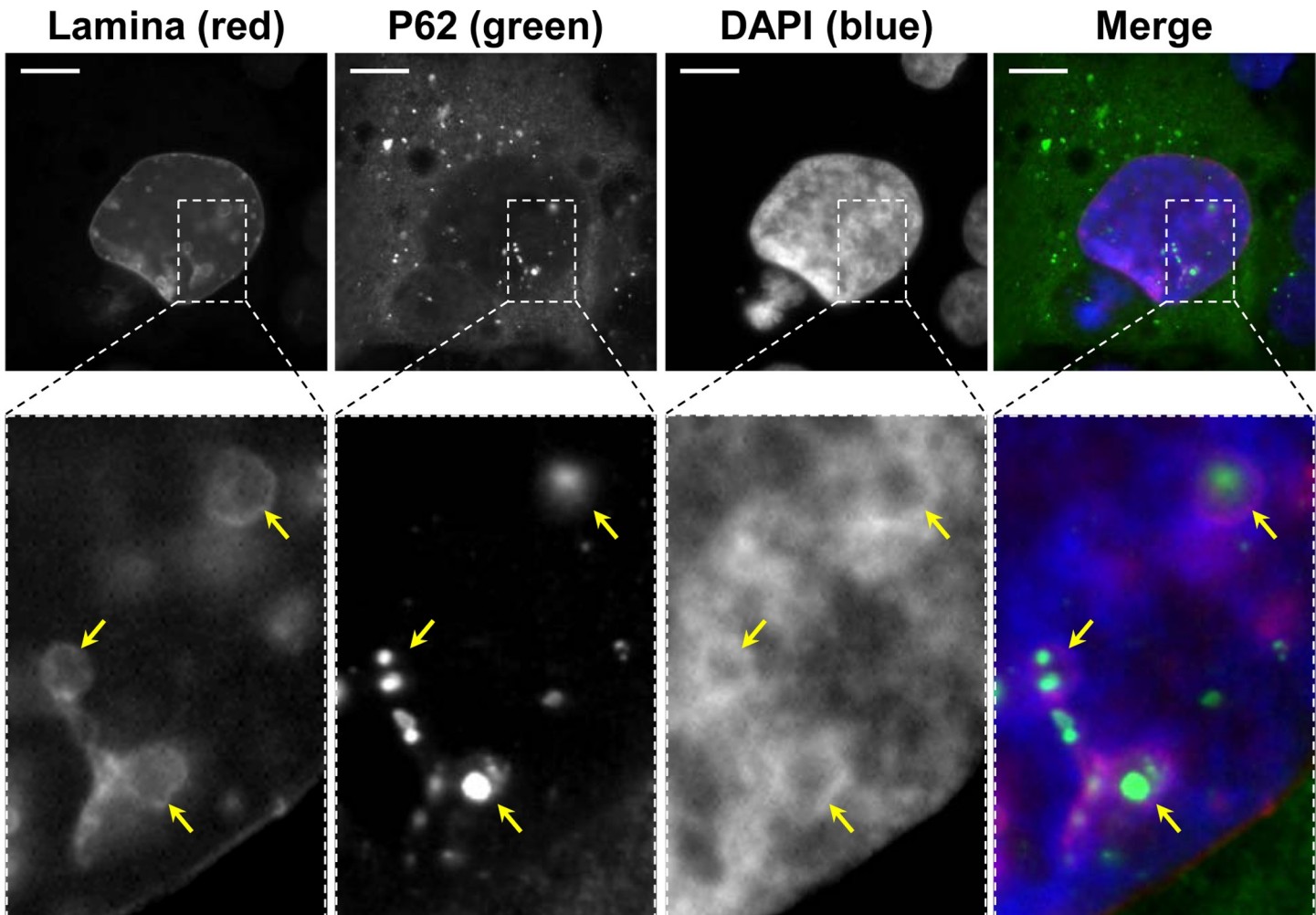

**Fig 5. Effect of *Helicobacter hepaticus* CdtB expression on nucleus remodeling.** Wide field image of Hep3B transgenic cells expressing the CdtB of *H. hepaticus* strain 3B1 processed for the nuclear lamina (red) and P62/SQSTM1 fluorescent staining (green), and DAPI to counterstain the nuclei (blue). Yellow arrows indicate DAPI-lacking nucleoplasmic reticulum enclosing P62/SQSTM1 bodies. Boxes correspond to enlargement. Scale bar, 10 μm. Abbreviations: DAPI, 4′, 6′-diamidino-2-phenylindol; P62, P62/SQSTM1.

as nucleoplasmic reticulum (NR) were observed both *in vivo* and in hepatic Hep3B and intestinal SW480 cell lines while tiny NR were more rarely observed in other cell lines as HT29 [6]. These phenotypes observed in Hep3B and HT29 cell lines are reminiscent of those observed for P62/SQSTM1 in the present study, suggesting that P62/SQSTM1 bodies could be a component of CDT-induced NR. NR invaginations originate from the nuclear envelope and are reversible and dynamic structures shown to be resorbed back into the envelope [22]. Their identification is thus carried out in tissues and cell culture by immuno-staining combined with microscopy [6]. The core of CDT-induced NR concentrates some RNA binding proteins of eIF4F complex and the subunits of the major coding-region determinant (mCRD)-mediated mRNA instability complex [6]. Thus, UNR/CSDE1 (upstream of N-Ras/Cold shock domain-containing protein E1), a subunit of the mCRD complex, was used to monitor the formation of NR using immunofluorescence microscopy [6] with a concomitant staining of P62/SQSTM1 and nuclei.

For this study, Hep3B and SW480 intestinal cells (S2D Fig) were used because NR formation is easily detected in these cell lines [6]. Both UNR protein and P62/SQSTM1 bodies were found randomly distributed in the cytosol upon CdtB intoxication. As reported, the nuclear remodeling induced by CdtB was associated with the formation of NR lacking DAPI staining and concentrating UNR-rich foci in giant nuclei. P62/SQSTM1 bodies were also invaginated in those CdtB-induced NR together or in the vicinity of UNR-rich foci (Fig 6A and 6B). In another cell line (SK-Hep-1), P62/SQSTM1 bodies were present in the cytosol colocalized with UNR-rich foci (S3A1 Fig) and in NR (S3A2 and S3A3 Fig). These effects were also observed *in vivo*. Indeed, immunofluorescent staining of the liver of mice infected with *H. hepaticus* for 14 months [23] confirmed the nuclear remodeling of hepatocytes in association with the formation of NR concentrating UNR-rich foci in giant nuclei, as previously reported [6]. P62/SQSTM1 bodies were also found invaginated in CdtB-induced NR of large hepatocytes (S3B Fig).

Whether there is CDT-induced NR formation or not, the size and location of P62/SQSTM1 varied in response to CDT/CdtB. Overall, all P62/SQSTM1 large bodies or aggregates were found near UNR-rich foci (Figs 6 and S3). Micronuclei-like structures surrounded by P62/SQSTM1 intense staining were observed (Fig 7A and 7B). Moreover, large P62/SQSTM1 aggregates were found near the distended nuclei (Fig 7C). These structures, which probably do not correspond to micronuclei, were strongly positive for phosphorylated H2A histone (γH2AX) and seemed to be surrounded with P62/SQSTM1 and/or LC3 (Figs 7C and 8A). Some P62/SQSTM1 large bodies/aggregates were also found tightly connected to the lamina of the nuclear membrane, and the nuclear lamina seemed to engulf these aggregates (Fig 8B). Some other structures with low DAPI staining were devoid of intense P62/SQSTM1 staining. All of these effects were not observed in non-infected cells and in the cells infected with *H. hepaticus* CdtB-H265L, lacking CdtB activity.

## CDT-associated cell survival involves autophagy and nucleoplasmic reticulum formation

CdtB-expressing cells were also treated with bafilomycin A1 and chloroquine with subsequent quantification of NR (Fig 9A). The number of CdtB-induced NR decreased significantly in presence of autophagy inhibitors, as compared to untreated cells, suggesting the involvement of autophagy in the formation of NR upon CdtB intoxication.

In order to better understand the effects of the inhibition of autophagy on the formation of NR, we studied the effect of silencing two key autophagy genes, ATG5 and ATG7. For this study, we used lentiviruses that express Cas9 along with single guide (sg) RNAs that target

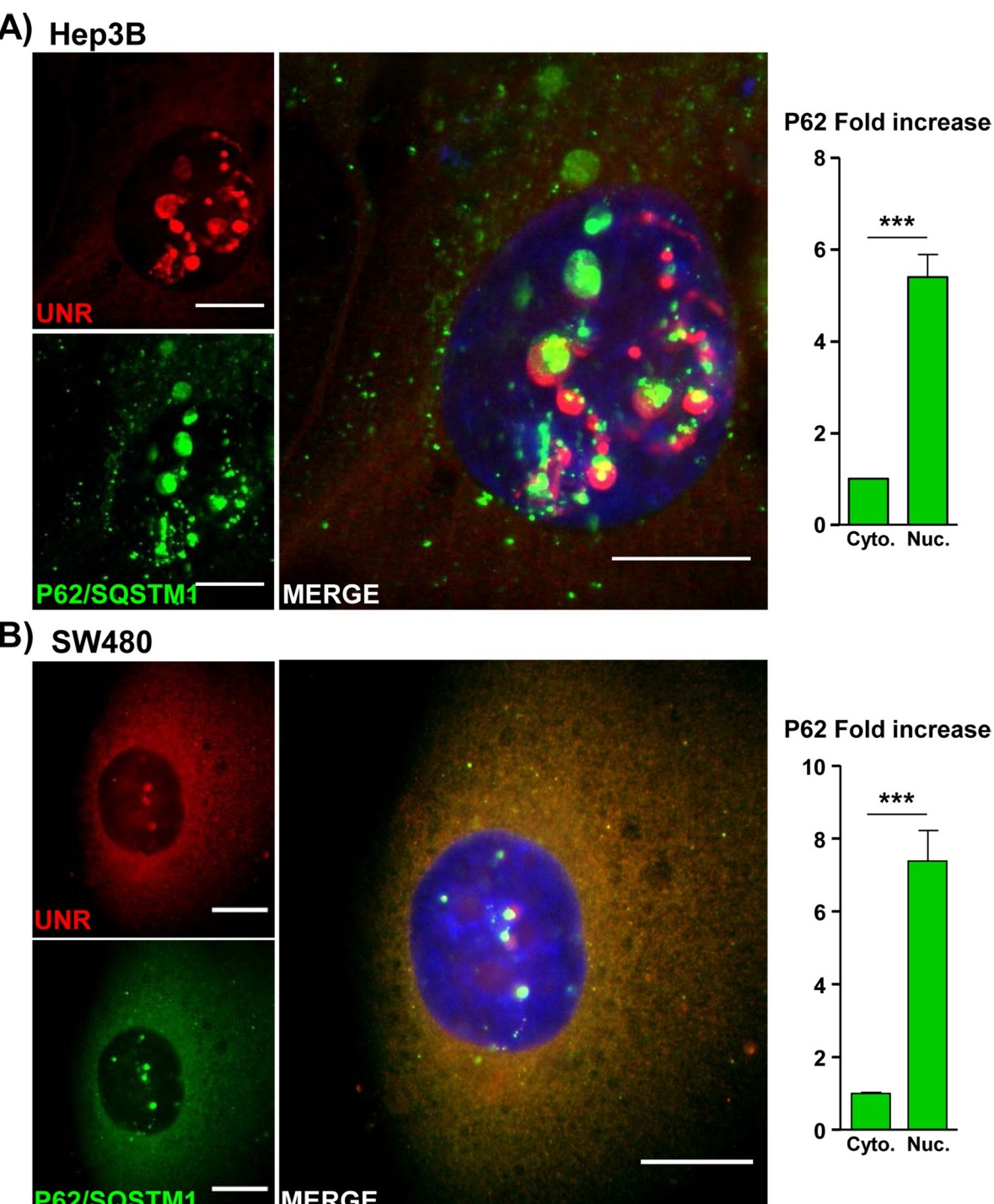

**Fig 6. Effects of *Helicobacter hepaticus* CdtB on P62/SQSTM1 localization.** As previously demonstrated, NR formation is primarily observed in response to CDT intoxication, *via* its active CdtB subunit [6]. Thus, images of non-infected cells are not presented below. **(A)** Hep3B and **(B)** SW480 Transgenic cells were cultivated with doxycycline for 72 h to induce the expression of the CdtB of *H. hepaticus* strain 3B1 [21]. Then cells were processed for staining with primary and fluorescent secondary antibodies: P62/SQSTM1 (red), UNR (green) and DAPI to counterstain the nuclei (blue). Subsequent quantification of P62/SQSTM1 in nucleoplasm, cytoplasm and foci were performed using capture of fluorescent staining (confocal imaging) by measuring the pixel intensity with the "Plot Profile" function of ImageJ (v. 1.52n), each count being performed on 100 NRs. The relative expression rate of P62/SQSTM1 in NR in response to the CdtB was reported as fold increase *versus* the expression in the cytosol. Scale bar, 20 μm. ***p<0.0001. Abbreviations: Cyto., cytoplasm; DAPI, 4′, 6′-diamidino-2-phenylindol; NR, nucleoplasmic reticulum; Nuc., nucleus; P62, P62/SQSTM1.

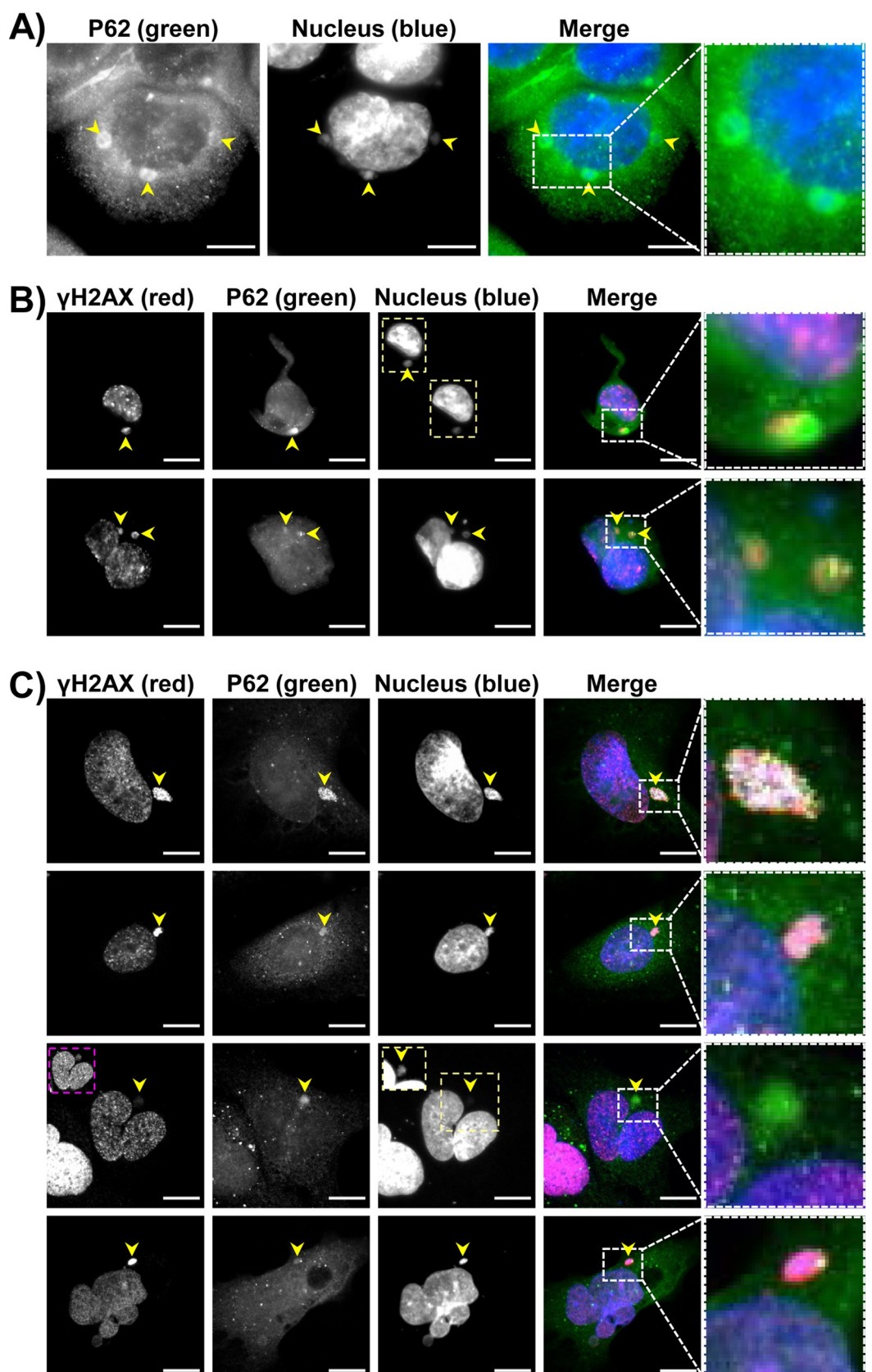

**Fig 7. Effects of bacterial genotoxins on P62/SQSTM1 bodies localization and γH2AX in intestinal and hepatic cell lines.** HT29 and Hep3B cells were infected for 3 days with CDT-secreting *H. hepaticus* or colibactin-secreting extra-intestinal pathogenic *E. coli*. Then, cells were processed for fluorescent staining with primary antibodies generated against γH2AX (red), P62/SQSTM1 (green), associated with fluorescent-labeled secondary antibodies and DAPI to counterstain the nuclei (blue). Fluorescent staining was observed using wide field fluorescence imaging. **(A)** Images of HT29 cells: P62/SQSTM1 (green), DAPI (blue). **(B)** Images of HT29 cells: γH2AX (red), P62/SQSTM1 (green), DAPI (blue). **(C)** Images of Hep3B cells: γH2AX (red), P62/SQSTM1 (green), DAPI (blue). Scale bars, 20 µm. Yellow arrowheads indicate extra-nuclear structures containing chromatin and P62/SQSTM1 aggregates. Yellow and pink boxes correspond to enlargement in which the DAPI and γH2AX were overexposed, respectively, in order to see the micronucleus-like structures. White boxes on the right correspond to enlargement of the merge. Abbreviations: DAPI, 4′, 6′-diamidino-2-phenylindol.

ATG5 or ATG7 or a non-specific sg RNA control (Mock) and then selected cells using puromycin [24]. RFP-, CdtB- and H265L-transgenic cell lines were previously selected using puromycin [21], so it was therefore not possible to use these cells. We thus performed this study on non-transgenic Hep3B cells.

A time-course coculture experiment was performed with Mock-, ATG5- and ATG7-KO Hep3B cells infected with *H. hepaticus* and its CDT corresponding ΔCDT strain and the effects of the infection following gene extinction were analyzed. As NR are transient and dynamic structures invaginated in the nucleus, their formation was analyzed using immunofluorescence microscopy. It should be noted that during co-culture experiments, not all cells are infected. Moreover, it was not possible to determine the percentage of infected cells, since no antibody targeting specifically *H. hepaticus* is available for immunofluorescence analysis. With regard to Mock-, ATG5- and ATG7-KO cells infected with the ΔCDT mutant strain, *H. hepaticus* infection led to profound nuclear remodeling with enlarged nuclei in association with an increase in γH2AX foci, a surrogate marker for double-stranded DNA breaks (Figs 9B and 9C and 10). CDT-induced nuclear reorganization was associated with the formation of UNR-rich cytoplasmic foci invaginated in the nucleus of giant cells (Figs 9D and 10). As previously reported [6], the stronger γH2AX signal correlated with the bigger nuclei.

These effects were associated with a decrease in cell number and an increase in the large fragment of caspase-3 resulting from cleavage adjacent to Asp175 indicative of caspase-3 activation, a feature of apoptotic cell death (Figs 9E and 9F and 10). All of these effects were concomitant with the increase in P62/SQSTM1 (Figs 9G and 10). With the exception of the cell proliferation curve (Fig 9E), the overall CDT-induced effects observed on Mock-Hep3B cells started to gradually increase until the third day after *H. hepaticus* infection to reach a maximum effect between the third and fifth day after infection; then a concomitant resumption of these effects started and the effects gradually decreased until the eighth day. None of these effects were observed in the Mock-KO cells infected with the ΔCDT mutant strain, showing that the CDT is the main virulence factor associated with these phenotypes. When compared to the Mock-KO cells infected with *H. hepaticus*, *H. hepaticus* infection of ATG5- and ATG7-KO cells led to narrower nuclear remodeling with less-distended nuclei, less γH2AX foci, and this was associated with the formation of less UNR-NR (Figs 9B–9D and 10). The decrease in these effects was concomitant with an important decrease in cell proliferation and an increase in apoptotic cells (Figs 9E and 9F and 10). With regards to P62/SQSTM1, a strong increase in the protein was observed in ATG5- and ATG7-KO infected cells in response to *H. hepaticus* infection, compared to the Mock-KO cells infected with *H. hepaticus* (Figs 9G and 10). P62/SQSTM1 increase was already present in ATG5- and ATG7-KO cells at the basal level due to the inhibition of autophagy flux.

The formation of NR was also observed in response to colibactin-induced DNA damage [6]. Similar experiments were thus carried out with those cell lines infected with colibactin-

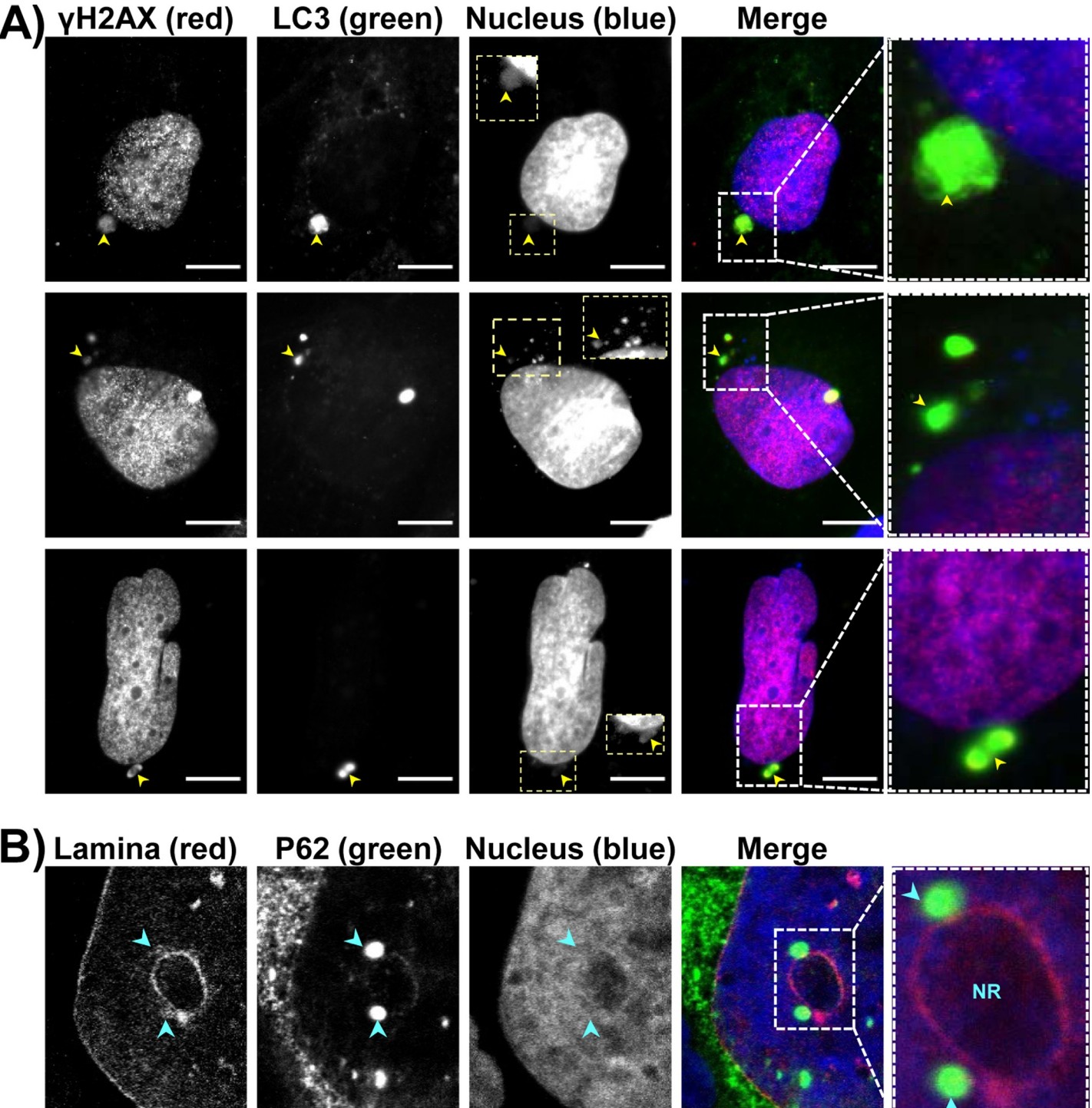

**Fig 8. Effects of bacterial genotoxins on P62/SQSTM1 bodies localization and LC3 in hepatic cell lines.** Hep3B cells were processed as in Fig 7. Fluorescent staining was observed using wide field fluorescence imaging. **(A)** Images of Hep3B cells: γH2AX (red), LC3 (green), DAPI (blue). **(B)** Images of Hep3B cells: nuclear lamina (red), P62/SQSTM1 (green), DAPI (blue). Scale bars, 20 μm. Yellow arrowheads indicate extra-nuclear structures containing chromatin. Blue arrowheads indicate P62/SQSTM1 aggregates tightly connected all along the nuclear membrane. Yellow boxes correspond to enlargement of micronucleus-like structures with DAPI. White boxes on the right correspond to enlargement of the merge. Abbreviations: DAPI, 4′, 6′-diamidino-2-phenylindol; NR, nucleoplasmic reticulum.

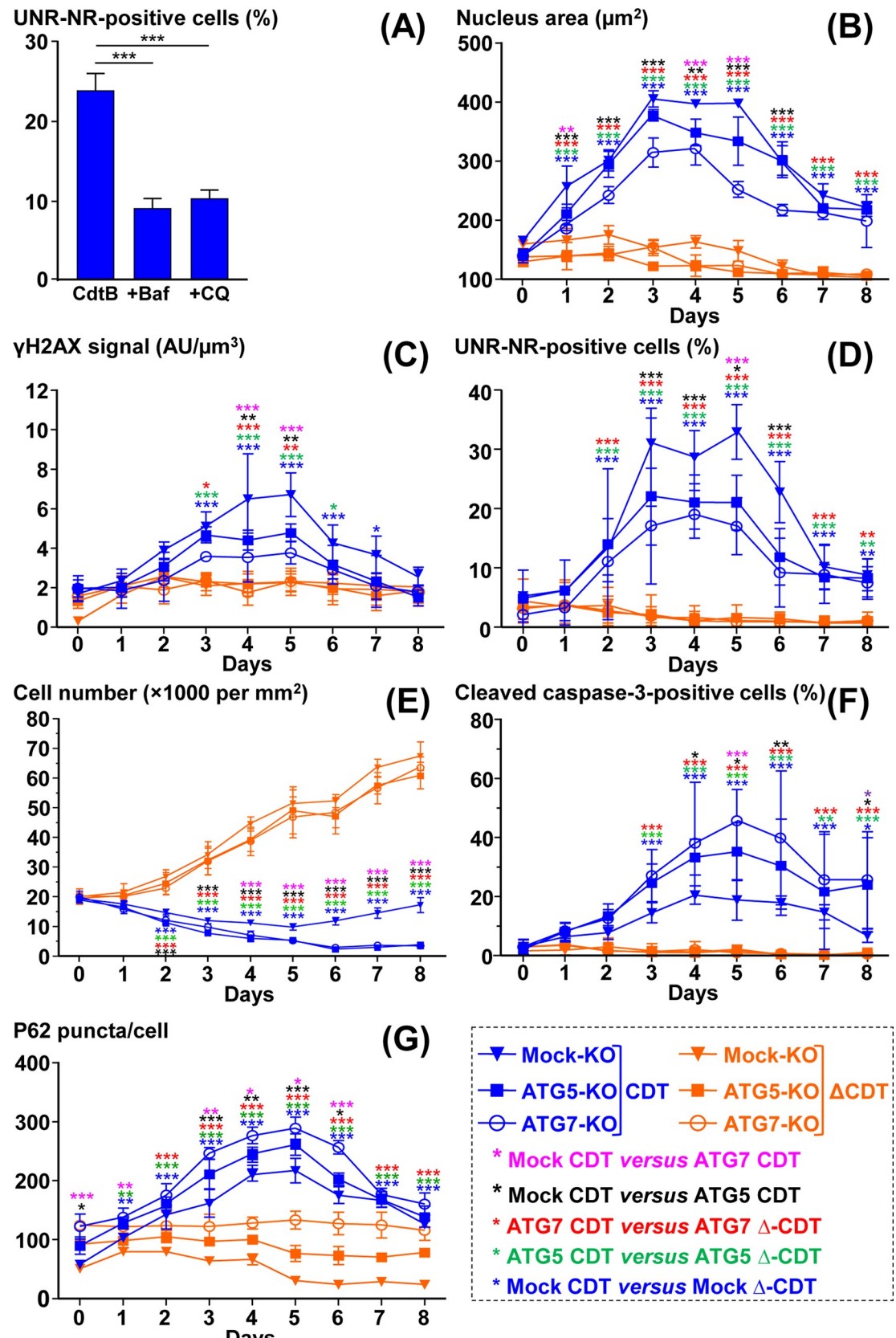

**Fig 9. Time-course analysis of the effects of ATG5 and ATG7 silencing on *Helicobacter hepaticus* CDT-induced effects. A)** CdtB-transgenic Hep3B cell line cultivated with doxycycline for 72 h to induce the expression of the CdtB of *H. hepaticus* strain 3B1. Cells were also treated with bafilomycin A1 (30 nM) or chloroquine (30 μM) 48 h and 24 h after doxycycline induction for a duration of 24 h and 48 h, respectively. Then cells were processed for fluorescent staining with primary antibodies generated against UNR (red) and P62/SQSTM1 (green) associated with fluorescent labeled-secondary antibodies and DAPI to counterstain the nuclei (blue). Quantification of UNR-NR positive cells (%) was performed on a minimum of 500 nuclei. The results are presented as the mean in one representative experiment (performed in triplicate) out of three. **B)** to **G)** Mock-KO, ATG5-KO and ATG7-KO Hep3B cells were infected for 3 days with *H. hepaticus* and its corresponding ΔCDT mutant strain. Then, the medium was removed, new medium was added and incubation continued until 8 days. These cells were processed daily for fluorescent staining with DAPI to detect the nucleus and fluorescent primary and secondary antibodies targeting γH2AX, UNR, cleaved caspase-3, and P62/SQSTM1. Fluorescent staining was observed using wide field fluorescence imaging (Fig 10). The results are presented as the mean in one representative experiment (performed in triplicate) out of three. A minimum of 500 cells were measured. **(B)** Nucleus surface (area) was quantified by isolating the DAPI fluorescence for each nucleus by using the 'Threshold' function of ImageJ (v. 1.52n). Nucleus size was measured in viable and early apoptotic cells. **(C)** γH2AX foci quantification was performed in viable and early apoptotic cells by measuring the pixel intensity with the "Integrated density" measure function of ImageJ (v. 1.52n). **(D)** The percentage of cells presenting UNR-NR was determined by manually counting the number of nuclei displaying UNR spots in the nucleoplasm. **(E)** Cell number quantification was performed manually. **(F)** Caspase-3-positive cells were quantified by counting the caspase 3 positive cells on 10 fields. $^*p<0.05$, $^{**}p<0.01$, $^{***}p<0.001$. Abbreviations: AU, arbitrary unit; Baf., bafilomycin A1; CQ, chloroquine; ΔCDT, CDT isogenic mutant of *H. hepaticus* strain 3B1; DAPI, 4′, 6′-diamidino-2-phenylindol; KO, Knock-Out, NR, nucleoplasmic reticulum; P62, P62/SQSTM1.

secreting extra-intestinal pathogenic *E. coli* (pks$^+$ *E. coli*). As expected, a profound nuclear remodeling was observed in response to colibactin in Mock-KO cells, compared to non-infected cells or cells infected with pks$^-$ *E. coli* (Fig 11A and 11C). This effect was associated with an increase in both LC3 puncta, γH2AX foci, NR formation, P62/SQSTM1 bodies, increased cell death and caspase-3 activity (Figs 11B and 11D and 12A–12D). Disruption of autophagy by silencing ATG5 and ATG7 inhibited the enlargement of the nuclei induced by colibactin while enhancing the caspase-3 activity and decreasing cell number, suggesting that autophagy protects cells against colibactin-induced apoptotic cell death. Disruption of autophagy led to an increase in colibactin-induced micronuclei-like structures. This increase was significant for ATG7-KO cells (Fig 12E) suggesting that micronuclei-like structures were not removed upon disruption of ATG7. These latest results suggest an autophagic-based selective removal of these micronuclei, also called nucleophagy. A decrease in cells presenting colocalized γH2AX foci and P62/SQSTM1 bodies was also observed in response to colibactin (Fig 12F) in ATG5- and ATG7-KO cells. This decrease likely reflects the global decrease in γH2AX foci upon autophagy disruption.

Taken together, all of these results showed that CDT and colibactin enhance autophagy leading to selective removal of genotoxin-induced micronuclei-like structures and protecting the cells against the genotoxin-induced apoptotic cell death. Pro-survival autophagy can also be associated with the formation of NR concentrating the autophagic receptor P62/SQSTM1 clustered in the genotoxin-induced nucleoplasmic reticulum.

## DNA damaging agents induce nuclear remodeling and autophagy

Both CDT and colibactin trigger DNA double-strand breaks. Colibactin was shown to alkylate DNA [25]. We therefore evaluated DNA damaging chemotherapeutic agents with different mechanisms of action: etoposide, a topoisomerase II enzyme inhibitor, and streptozotocin, a glucosamine-nitrosourea alkylating compound (Fig 13). Both compounds trigger NR formation enclosing UNR and P62/SQSTM1 bodies as well as autophagy activation. Moreover, DAPI-lacking large aggregates positive for LC3 and γH2AX foci were also found. Thus, NR formation and autophagy are not restricted to CDT and colibactin, but occur also in response to other non-alkylating and alkylating DNA damaging agents.

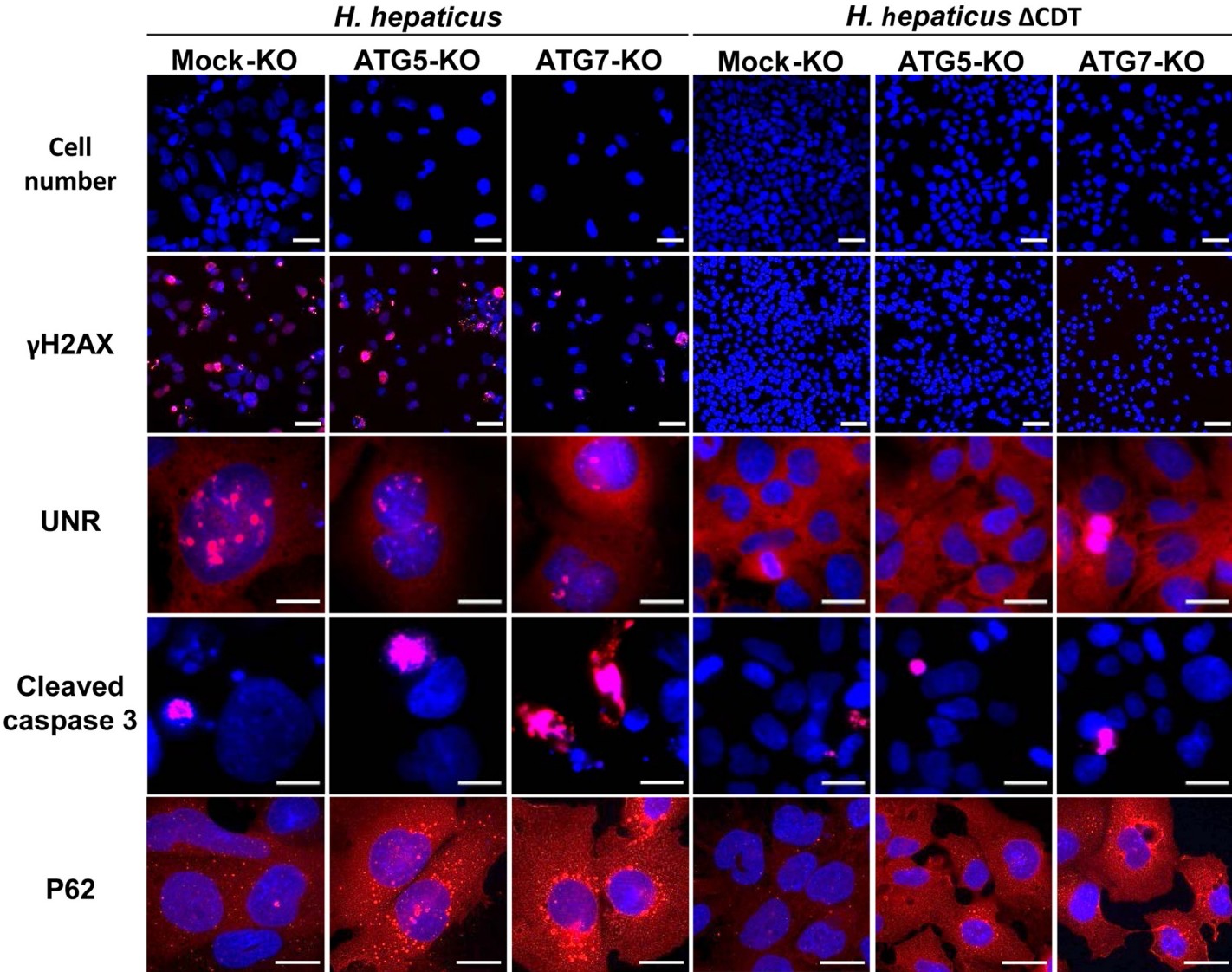

**Fig 10. Images of the effects of ATG5 and ATG7 silencing on Helicobacter hepaticus CDT-induced effects.** (Fig 9 continued) Mock-KO, ATG5-KO and ATG7-KO Hep3B cells were processed as in Fig 9B–9G. Cells were stained for nuclei (DAPI), as well as for γH2AX, UNR, cleaved caspase-3, and P62/SQSTM1 (red), along with DAPI to counterstain the nuclei (blue). Wide field images of cocultures experiment at days 4 are presented. Scale bars: 50 μm for cell number (DAPI only), 100 μm for γH2AX and DAPI double staining, and 20 μm for other double staining (UNR, cleaved caspase-3, and P62/SQSTM1 with DAPI). *p<0.05, **p<0.01, ***p<0.001. Abbreviations: ΔCDT, CDT isogenic mutant of *H. hepaticus* strain 3B1; DAPI, 4′, 6′-diamidino-2-phenylindol; KO, Knock-Out, P62, P62/SQSTM1.

## Discussion

In the present study, we showed that treatment of cells with the bacterial genotoxin CDT/CdtB induced autophagy. Whole genome microarray data performed on intestinal cells pointed to numerous mRNA upregulated in response to CdtB intoxication. Little data relative to the regulation of autophagy-related genes in response to CDT are known [3]. However, it was shown that patient colonic mucosa colonized with colibactin-producing *E. coli* presented higher levels of transcript encoding proteins involved in autophagy than colonic mucosa colonized with *E. coli* that do not carry the pks island [5]. Most of these transcripts were also found upregulated by CdtB in the present study (ATG5, ATG12, ATG13, ATG16L1, BECN1, MAP1LC3A/B, WIPI1).

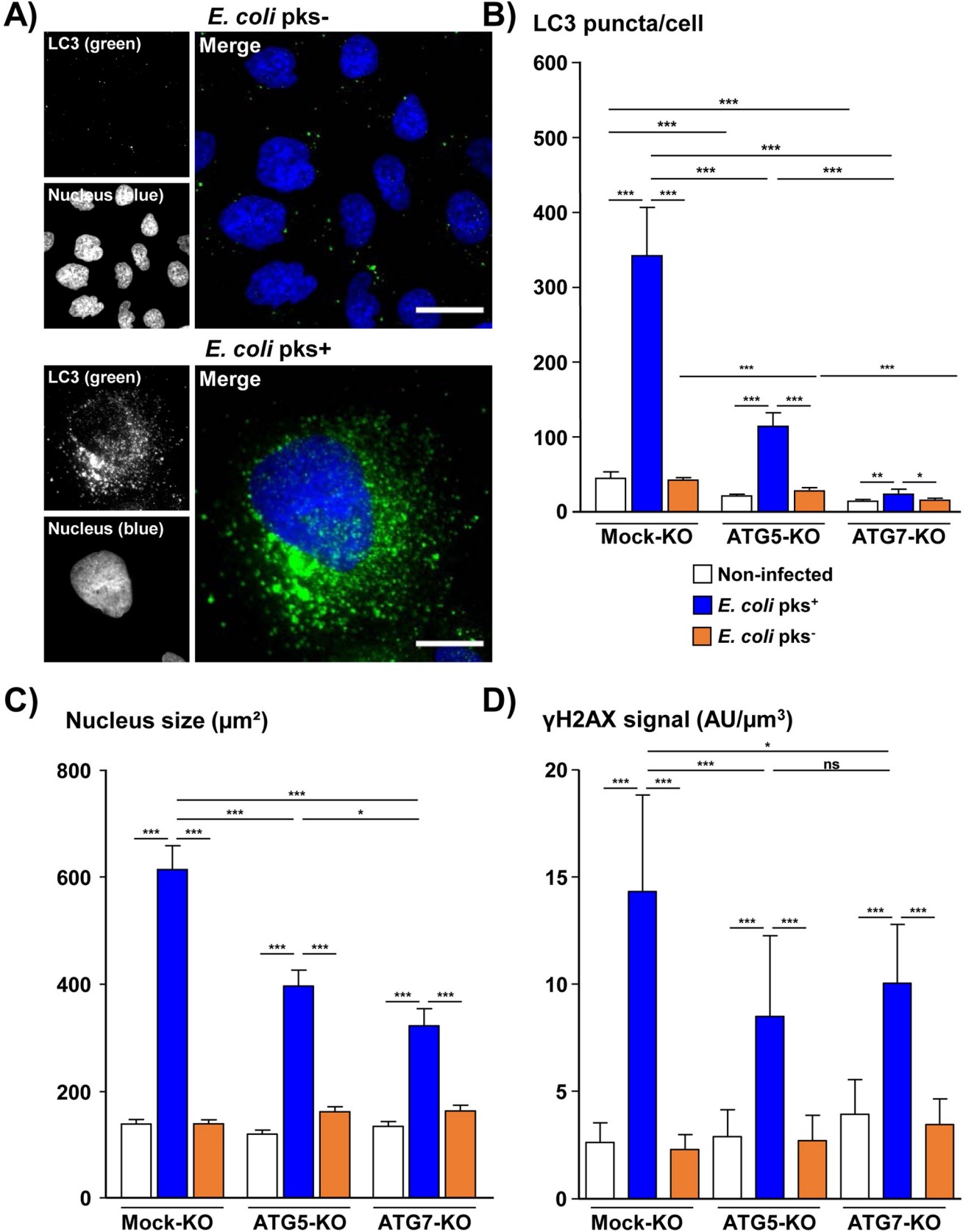

**Fig 11. Analysis of the effects of ATG5 and ATG7 silencing on *Escherichia coli* colibactin-induced effects.** Mock-KO, ATG5-KO and ATG7-KO Hep3B cells were infected for 4 hours with colibactin-secreting extra-intestinal pathogenic *E. coli* and its corresponding isogenic mutant and cultivated for 3 days in a bacteria free medium. Then, cells were processed for fluorescent staining with primary antibodies generated against LC3 and γH2AX associated with fluorescent labeled-secondary antibodies (green) and DAPI to counterstain the nuclei (blue). **(A)** Wide field images of Hep3B cells: LC3 (green), DAPI (blue). Scale bar, 20 μm. **(B)** The number of fluorescent LC3 puncta was quantified using the "Find Maxima" function of ImageJ. **(C)** Nucleus surface (area) was quantified in viable and early apoptotic cells by isolating the DAPI fluorescence for each nucleus by using the 'Threshold' function of ImageJ (v. 1.52n). **(D)** γH2AX foci quantification was performed in viable and early apoptotic cells by measuring the pixel intensity with the "Integrated density" measure function of ImageJ (v. 1.52n). A minimum of 500 cells were measured. *$p<0.05$, **$p<0.01$, ***$p<0.001$. Abbreviations: AU, arbitrary unit; DAPI, 4′, 6′-diamidino-2-phenylindol; KO, Knock-Out, pks-, bacterial artificial chromosome vector; pks+, bacterial artificial chromosome vector with pks island encoding colibactin.

We found that the autophagy flux is activated in cells subjected to CdtB, as previously reported [4]. Seiwert *et al.* also showed that CDT-induced DSBs trigger pro-survival autophagy, as demonstrated during experiments with chloroquine [4]. Similarly, pharmacological inhibitors of autophagy (bafilomycin A1, chloroquine) and CRISPR-Cas9 silencing of ATG5 and ATG7 genes increased colibactin- and CDT-induced apoptotic cell death, which was not observed during transient silencing of ATG5 [4]. In the present study, *H. hepaticus* CDT and *E. coli* colibactin are most likely the main virulence factors associated with the induction of autophagy since the corresponding mutants (ΔCDT and pks-) play a negligible role in this effect, in agreement with previous data [4,5]. Autophagy induced by bacteria degrades internalized pathogens in addition to the infected cell and reduces the spread of infection. For CDT-secreting *H. hepaticus* and colibactin-secreting *E. coli*, their genotoxin would be the main virulence factor responsible for the induction of autophagy and bacterial clearance. In addition, *H. hepaticus* CdtB also regulated some genes involved in the 'apoptosis and autophagy' pathway and the increase in the mRNA level of apoptosis-regulator proteins and inflammatory caspases suggests CdtB-induced pyroptosis during *H. hepaticus* infection. Both autophagy and apoptosis are thus induced following CDT and colibactin intoxication. This is in agreement with some studies, which showed that the regulation of apoptosis and autophagy is intimately connected and that both of these supposedly different processes, can be stimulated by the same stresses [26]. Under certain circumstances, autophagy can protect cells from various apoptotic stimuli by preventing them from undergoing apoptosis, whereas in other cellular settings, autophagy is an alternative cell-death pathway [26]. Accordingly, we can assume that autophagy constitutes an adaptation to the genotoxic stress induced by CDT and colibactin, which counteracts the genotoxin-induced apoptosis to maintain cell survival during DDR. However, for cells whose DNA damage is beyond repair, autophagy might not be sufficient to maintain cell survival leading to apoptotic cell death. In the present study, γH2AX foci and nucleus size were measured in viable and early apoptotic cells; thus, their global decrease in response to CDT and colibactin upon autophagy disruption did not reflect their overall cellular quantification but likely reflects the enhanced apoptotic cell death. Moreover, it is also possible that autophagy disruption conducted to activation of alternative DDR pathways (instead of the canonical pathway) [27], which functioned as backup and processed the genotoxin-induced DNA breaks, resulting in γH2AX foci and nucleus size decrease. Further investigation is needed to explore the DDR molecular regulation in autophagy-deficient cells in the context of bacterial genotoxin response. Preliminary results showed that autophagy disruption by ATG7 silencing led to increased IL8 secretion and nuclear factor κB translocation following bacterial genotoxin intoxication (S4A and S4B Fig), suggesting that autophagy would protect intoxicated cells by reducing inflammatory responses. However, these latest results deserve further study.

We found that CDT/CdtB increased the expression levels of both P62/SQSTM1 mRNA and protein, in contrast to previous experiments with purified CDT [4]. The difference between

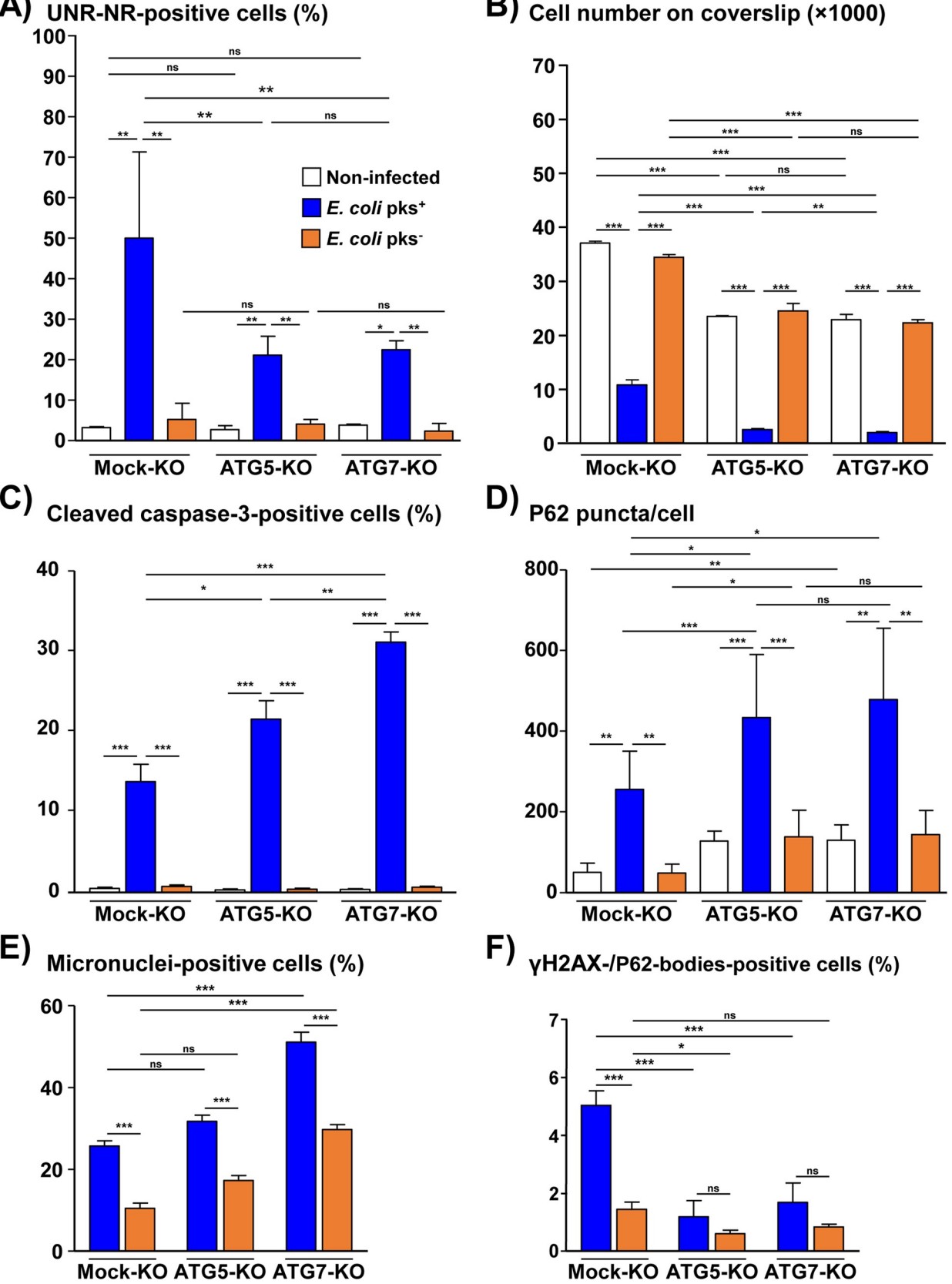

**Fig 12. Analysis of the effects of ATG5 and ATG7 silencing on *Escherichia coli* colibactin-induced effects (Fig 11 *continued*) Mock-KO, ATG5-KO and ATG7-KO Hep3B cells were processed as in Fig 11.** Then, cells were processed for fluorescent staining with primary antibodies generated against UNR, cleaved Caspase-3 or P62/SQSTM1 or γH2AX associated with fluorescent labeled-secondary antibodies and DAPI to counterstain the nuclei. **(A)** The percentage of cells presenting UNR-NR was determined by manually counting the number of nuclei displaying UNR spots in the nucleoplasm. **(B)** Cell number quantification was performed manually. **(C)** Caspase-3-positive cells were quantified by counting the caspase 3 positive cells on 10 fields. **(D)** P62/SQSTM1 bodies were quantified using the "Find Maxima" function of ImageJ. The results are presented as the mean in one representative experiment (performed in triplicate) out of three. **(E)** Quantification of micronucleus-like structures in Hep3B cells was performed manually. **(F)** Quantification of Hep3B cells with colocalized γH2AX-foci and P62/SQSTM1-bodies was performed manually. A minimum of 500 cells were measured. *p<0.05, **p<0.01, ***p<0.001. Abbreviations: AU, arbitrary unit; DAPI, 4', 6'-diamidino-2-phenylindol; KO, Knock-Out, NR, nucleoplasmic reticulum; P62, P62/SQSTM1; pks⁻, bacterial artificial chromosome vector; pks⁺, bacterial artificial chromosome vector with pks island encoding colibactin.

P62/SQSTM1 expression most likely reflects the amount of internalized CdtB in the cell. The expression of the P62/SQSTM1 protein level cannot be used as an indicator of the autophagy activity as SQSTM1 mRNA level is often upregulated under stressful conditions [28]. Thus, the important increase in P62/SQSTM1 mRNA by CdtB, most likely hides the degradation of P62/SQSTM1 protein by autophagy [29,30]. The significant increase in the P62/SQSTM1 phosphorylated (Ser403) isoform induced by CdtB suggests selective autophagic clearance of ubiquitinated proteins and protein aggregates that are poorly degraded by proteasomes.

Activation of prosurvival autophagy requires clusterin expression and this cytoprotective chaperone facilitates stress-induced lipidation of LC3 and induces autophagosome biogenesis [31]. Clusterin protects against genotoxic stress and suppresses DNA damage-induced cell death [32]. Clusterin-enhanced cell survival to DNA damage occurs therefore *via* autophagy. We previously reported that clusterin mRNA is highly upregulated (~8-fold increase) in response to *H. hepaticus* CdtB [33]. In line with that, clusterin may be involved in pro-survival autophagy following DNA damage triggered by CdtB.

DNA damage following exposure to bacterial toxins induces cell death (necrosis, apoptosis) and cellular senescence [21,34,35]. Senescent cells following DNA damage display multiple morphological aspects such as cytoskeleton and organelle remodeling and nuclear size enlargement. Selective degradation of the entire nucleus or nuclear components occur in mammals in order to maintain nuclear integrity. Exposure to CDT and colibactin induces the formation of nucleoplasmic bridges and micronuclei [34,36]. Micronuclei contain damaged chromosome fragments enclosed by the nuclear membrane that can be sequestered and degraded by autophagy [37]. Indeed, 2 to 5% of micronuclei induced by cell cycle blocker compounds 1) show a decreased intensity in DAPI staining consistent with chromatin degradation in these compartments, 2) exhibit γH2AX-positive DNA damage foci, 3) colocalize to LC3-positive vesicles and 4) contain P62/SQSTM1 [37]. In the context of bacterial genotoxin, micronuclei-like structures with DAPI, γH2AX, LC3 and P62/SQSTM1 were observed, suggesting micronuclear autophagy. Regardless of the P53 status of the line used, similar results were observed. Indeed, *TP53* gene is mutated in HT29 and Hep3B [38,39] resulting in a high level of constitutive P53 protein expression in HT29 [21] and a truncated protein in Hep3B while no mutations occurred in SK-Hep-1.

Senescent cells secrete the senescence-associated secretory phenotype (SASP) and upregulate the formation and release of small extracellular vesicles (exosomes, microvesicles, nucleosomes and apoptotic bodies) from epithelial cells [40]. Small extracellular vesicles and autophagy cooperatively prevent apoptotic cell death by removing cytoplasmic DNA fragments derived from chromosomal DNA or bacterial infections in order to protect cells from excessive inflammatory responses [40]. P62/SQSTM1 is present within these vesicles [41]. P62/SQSTM1 bodies induced by CDT vary greatly in size and suggest the presence of different structures or agglomerates. Because inhibition of autophagy increases apoptotic cell death

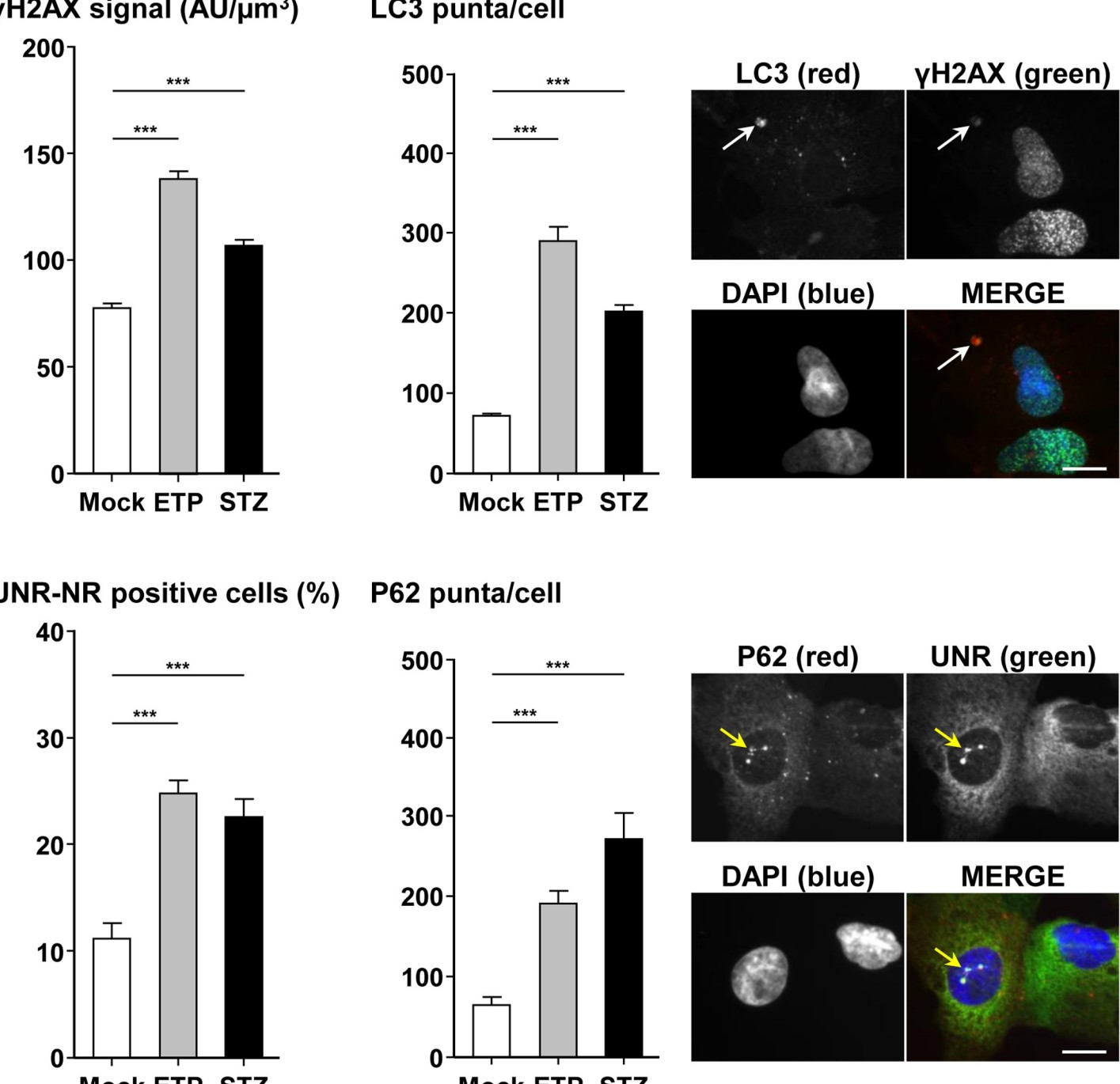

**Fig 13. Effects of DNA damaging agents on nuclear remodeling and autophagy.** Hep3B cells were cultivated in the presence of etoposide (5 µM) or steptozocin (10 mM) for a duration of 24 h. Then, the medium was removed and incubation was continued for 48 h. The cells were then stained with fluorescent primary and secondary antibodies targeting LC3 (red)/γH2AX (green), P62/SQSTM1 (red)/UNR (green) and DAPI to counterstain the nuclei (blue). Fluorescent staining was observed using wide field fluorescence imaging. γH2AX foci quantification was performed by measuring the pixel intensity with the "Integrated density" measure function of ImageJ (v. 1.52n). LC3 puncta and P62/SQSTM1 bodies were quantified using the "Find Maxima" function of ImageJ. The percentage of cells presenting UNR-NR was determined by manually counting the number of nuclei displaying UNR spots in the nucleoplasm. Quantification was performed on a minimum of 500 cells. The results are presented as the mean in one representative experiment (performed in triplicate) out of three. ***p<0.001. Scale bar, 20 µm. White arrow indicates DAPI-lacking large aggregate positive for LC3 and γH2AX foci. Yellow arrow indicates DAPI-lacking nucleoplasmic reticulum enclosing UNR and P62/SQSTM1 bodies. Abbreviations: AU, arbitrary unit; DAPI, 4′, 6′-diamidino-2-phenylindol; ETP, etoposide; NR, nucleoplasmic reticulum; P62, P62/SQSTM1; STZ, steptozocin.

upon CDT intoxication (Figs 9,10, and 12), some of these large CDT-induced P62/SQSTM1 bodies found in the cytosol could correspond to P62-positive extracellular vesicles prior to their externalization.

Senescent cells process their chromatin *via* the autophagy/lysosomal pathway, which contributes to the proteolytic processing of histones [42]. In this context, cytoplasmic chromatin fragments are closely juxtaposed to P62/SQSTM1 bodies in senescent cells [42]. It is believed that P62/SQSTM1 bodies enhance the nucleocytoplasmic export of chromatin in the cytoplasm of senescent cells [42]. A cytoplasmic access *via* CDT-induced NR would facilitate chromatin transport to the cytoplasm for degradation by autophagy pathway.

In addition to maintaining nuclear DNA homeostasis, nuclear autophagy (also known as nucleophagy) plays an important role in the post-transcriptional RNA modification process [43]. We previously reported that the nuclear remodeling induced by CDT *via* CdtB is associated with the formation of NR whose core concentrates mRNA and RNA binding proteins involved in mRNA translation and decay, amongst which is the cytoplasmic RNA-binding protein UNR/CSDE1 [6]. P62/SQSTM1 large bodies are also invaginated in the core of CdtB-induced NR together or in the vicinity of UNR-rich foci. UNR/CSDE1 influences proliferation under stress conditions, is overexpressed in melanoma tumors and promotes invasion and metastasis partly by inducing translation elongation of targeted mRNAs [44]. In melanoma, P62/SQSTM1 was shown to 1) extend mRNA half-life of a several pro-metastatic factors by opposing mRNA decay and 2) recruit RNA-binding proteins that are enriched within P62/SQSTM1 interactome [45]. In this context, P62/SQSTM1 would have recruited RNA-binding proteins from eIF4F and mCRD complexes, as UNR/CSDE1, to shuttle them in the core of CDT-induced NR, protecting and translating mRNA in order to maintain cell survival. The fact that P62/SQSTM1 bodies are systematically present near UNR-rich foci in response to CDT/CdtB (with or without formation of NR structure) supports this hypothesis.

P62/SQSTM1 was recently shown to be a RNA-binding protein whose vault RNAs are the major interacting RNA [46]. Vaults are large ribonucleoprotein particles composed of the major vault protein (VMP), two minor vault proteins (VPARP and TEP1) and a variety of small untranslated RNA molecules [47,48]. DNA-damaging agents directly activate the human *VMP* gene [49], but no increase in MVP transcript in response to the CdtB of *H. hepaticus* was observed in HT29 cells (microarray data). Given the association with the nuclear membrane and the location within the cell, vaults are thought to play roles in intracellular and nucleocytoplasmic transport processes. If vault ribonucleoprotein particles are involved in CDT/CdtB-induced NR, they may regulate nucleocytoplasmic transport in those structures. Vault RNA1-1 directly binds P62/SQSTM1 and inhibits P62/SQSTM1-dependent autophagy and ubiquitin aggregate clearance by interfering with P62/SQSTM1 multimerization [46]. CDT/CdtB induced-NR are deeply invaginated in the nucleus and concentrate large P62/SQSTM1-rich aggregates, most likely reflecting the absence of the small non-coding vault RNA1-1 in those protected structures in which P62/SQSTM1-dependent autophagy takes place.

P62/SQSTM1 contains two nuclear localization signals and a nuclear export signal. Thus, it shuttles continuously between nuclear and cytosolic compartments at a high rate and cytoplasmic P62/SQSTM1 body aggregates regulate its nucleocytoplasmic shuttling [19]. P62/SQSTM1 is also involved in the assembly of proteasome-containing degradative compartments in the vicinity of nuclear aggregates [19]. In this hypothesis, the bacterial genotoxins induce the transport of P62/SQSTM1 protein in the nucleus; this transfer would have been very fast because overall P62/SQSTM1 has not been visualized in the nucleus of the various models used in the present study.

Following various stresses, cytoplasmic ribonucleoprotein granules, *i.e.* stress granules (SG) and processing bodies (GW/P-bodies), are induced to protect untranslated mRNAs until stress

relief. SG and GW/P-bodies were not observed upon CDT intoxication [6] but the composition of these organelles is reminiscent of that of CdtB-induced NR [6]. As SG and GW/P-bodies are cleared from cells through autophagy [50], P62/SQSTM1 bodies may also participate in the clearance of messenger ribonucleoprotein particles clustered in CDT-induced NR structures.

Thus, many observations suggest a central role for P62/SQSTM1 upon infection with genotoxin- secreting bacteria in regulating DNA damage response. However, it should be noted that P62/SQSTM1 is not necessary for NR formation in this context, as P62/SQSTM1 silencing (CRISPR-Cas9 System) did not affect NR formation (S5A Fig) neither UNR expression. On the other hand, MAFB silencing [33] reduced NR formation and P62/SQSTM1 bodies increased during bacterial cocultures with *H. hepaticus* (S5B Fig). This effect of MAFB is not surprising as this oncoprotein is involved in nuclear remodeling following DNA damage induced by Helicobacter CDT [33].

Finally, cell survival following the DNA damage induced by CDT and colibactin involved prosurvival autophagy in intestinal and hepatic cells. Macroautophagy including nucleophagy occurred upon bacterial genotoxin intoxication and P62/SQSTM1 bodies may play a central role in this pro-survival autophagy, probably *via* different mechanisms in light of the observed disparity of P62/SQSTM1 bodies (shape, localization). Further studies are required to decipher the role of P62/SQSTM1 in the survival of cells stressed by bacterial genotoxins and more particularly its role in the formation of NR in light of its RNA-binding protein function recently demonstrated [46].

It should be emphasized that similar effects were also observed with DNA damaging agents used during chemotherapies, etoposide and streptozotocin, suggesting that NR formation and increased autophagy following DNA damage are not a specific adaptation to bacterial genotoxins but rather a survival mechanism following a genotoxic stress. Accordingly, these survival mechanisms may contribute to the resistance of cancer cells to therapies inducing DNA damage. In line with that, altered nuclear morphology reminiscent of NR was reported as a common feature of many cancers and has an adverse prognostic significance for various cancers [51]. These latest data should open new fields of investigation.

## Material & methods

### Ethics statement

Animal material provided from previous studies was approved by the Ethics Committee for Animal Care and Experimentation CEEA 50 in Bordeaux (Comité d'Ethique en matière d'Expérimentation Animale agréé par le ministre chargé de la Recherche, "dossier no. Dir 13126BV2", "saisines" no. 4808-CA-I [23] and 13126B [21], Bordeaux, France), according to treaty no. 123 of the European Convention for the Protection of Vertebrate Animals. Animal experiments were performed in an A2 animal facility (security level 2) by trained authorized personnel only.

### Transcriptomics

The global analysis of the expression of human genes in response to the *H. hepaticus* CdtB was quantified using the Human GE 4x44K v2 Microarray Kit (Agilent Technologies, Les Ulis, France). Total RNA was extracted with the miRNeasy Mini Kit (Qiagen) according to the manufacturer's recommendations. Cyanine-3 (Cy3) labeled cRNA was prepared from 0.2 μg RNA using the One-Color Microarray-Based Gene Expression Analysis kit (Low Input Quick Amp Labeling, Agilent Technologies) according to the manufacturer's instructions, followed by RNAeasy column purification (Qiagen). Dye incorporation and cRNA yield were checked

with a NanoDrop 2000 spectrophotometer. For each sample, a total of 1.65 μg of cRNA was fragmented and hybridized overnight at 65°C onto Human Gene Expression v2 4x44k Microarray (Agilent Technologies). The slides were washed as recommended by the manufacturer, and scanned on an Agilent G2565CA scanner, at a 5 micron resolution and using the 20-bit scan mode (Agilent Technologies). Images were processed with Feature extraction (version 10.7). For satistical and bioinformatic analyses, data were normalized for inter-array comparisons and analyzed using the BRB-ArrayTools package (version 4.2.0, http://linus.nci.nih.gov/BRB-ArrayTools.html). As recommended by Agilent Technologies, we chose the percentile method for the normalization, and adjusted the 75th percentile of all of the non-control probes to a value of 500. We identified genes that were expressed differentially among the two groups using a random-variance t-test (Class comparison between groups of arrays package, BRB-ArrayTools). Gene expression differences were considered statistically significant if they showed ≥30% difference in mean expression levels between samples from CdtB versus control conditions (tdTomato fluorescent protein), and if their associated p-value was <0.01.

## Reagents and antibodies

Bafilomycin A1 (#B1793), chloroquine (#C6628), doxycycline hyclate (#D9891), etoposide (#E1383), gentamicin (#G1522), puromycin (#P7255) and streptozocin (#S0130) came from Sigma Aldrich (Saint-Quentin Fallavier, France).

Antibodies used for western blot analyses were: mouse monoclonal anti-LC3 (clone 4E12) (#M152-3, MBL Life science) distributed by CliniSciences (Nanterre, France); mouse monoclonal anti-P62/SQSTM1 (Clone 3/P62 Ick ligand) (#610832) from BD Biosciences (San Jose, CA); rat monoclonal anti-phospho-P62/SQSTM1 (Ser403) (clone 4F6) (#MABC186, Merck Chimie Fontenay sous Bois), rabbit polyclonal anti-UNR/CSDE1 (#HPA018846) from Sigma Aldrich and mouse monoclonal anti-α-tubulin (clone DM1A) (#05–829) from Sigma Aldrich; rabbit polyclonal anti-AMPKα1 (#2795) and rabbit monoclonal anti-phospho-AMPKα (Thr172) (clone 40H9) (#2535) from Cell Signaling Technology (distributed by Ozyme, Saint-Cyr-L'École, France).

Antibodies used for immunofluorescence analyses were: mouse monoclonal anti-LC3 (clone 4E12) (#M152-3) (MBL Life Sciences) and mouse monoclonal anti-P62/SQSTM1 (D-3) (#sc-28359, Santa Cruz Biotechnology, Heidelberg, Germany) (1/100). The working conditions for anti-LC3 were 1/100 in Phosphate-Buffered Saline with 1% digitonin as permeabilizing agent. The simultaneous immunodetection of LC3 and γH2AX was performed under LC3 conditions. Rabbit polyclonal anti-UNR/CSDE1 (#HPA018846) (1/100) came from Sigma Aldrich. Rabbit monoclonal cleaved Caspase-3 (Asp175) (clone 5A1E) (1/200) (#9664), rabbit monoclonal anti-Caspase-1 (clone D7F10) (#3866) (1/100) and rabbit monoclonal anti-phospho-histone H2A.X (γH2AX) (Ser139) (20E3) (#9718) (1/100) came from Cell Signaling Technology. The antibody used for Caspase-1 immunodetection did not work for immunocytochemistry but gave a nice immunodetection of Caspase-1 on tissue sections. Rabbit polyclonal anti-53BP1 (#NB100-304 1/100) came from Novus Biologicals (Centennial, CO, USA). Rabbit polyclonal anti-NF-kB p65 (C-20) (#sc-372) (1/200) came from Santa Cruz Biotechnology (Santa Cruz, California, USA). Alexa 488 or Alexa 594-labeled secondary antibodies and 4',6'-diamidino-2-phenylindol (DAPI) were purchased from Molecular Probes (Invitrogen, Cergy Pontoise, France).

## Sequential 53BP1 and γH2AX immunostaining on tissue sections

Both anti-53BP1 and anti-γH2AX originated from rabbits. To correlate 53BP1 association with γH2AX, sequential fluorescent labeling was performed using 3-μm tissue sections, prepared from formalin-fixed paraffin-embedded human CdtB-Hep-3B xenografts. Firstly, 53BP1 immunodetection was performed, the stained tissue section was scanned, and the

coordinates of the scanned area were recorded to ensure repositioning of the slide after γH2AX labeling. Secondly, γH2AX immunodetection was performed using the same slide; the tissue section was scanned again, using the repositioning function of the microscope (Zeiss Axioplan 2 fluorescence microscope, Zeiss, Jena, Germany). As a result (S6 Fig), CdtB activated DNA damage signaling, detected as 53BP1/γH2AX-positive foci, showing that CdtB intoxication results in γH2AX phosphorylation and 53BP1 recruitment, as previously reviewed in [35].

## Cell lines

The human transgenic HT29 (DSMZ no. ACC 299) and SW480 (ATCC CCL-228) cell lines were derived from a colon adenocarcinoma, and the human epithelial cell line Hep3B (ATCC HB-8064) from a hepatocellular carcinoma. The corresponding transgenic cell lines were established as previously reported [6,21] and grown in their respective culture medium supplemented with 10% heat-inactivated fetal calf serum (Invitrogen) at 37°C in a 5% $CO_2$ humidified atmosphere. When required, the transgene expression was induced in the cells from the tetracycline-inducible promoter by addition of doxycycline (200 ng/ml) to the culture medium and incubation for 72 h. In some experiments, cell lines were also treated with bafilomycin A1 (30 nM) or chloroquine (30 μM) to inhibit the autophagy process. Bafilomycin A1 and chloroquine were added 48 h and 24 h after doxycycline induction for a duration of 24 h and 48 h, respectively.

## ATG5, ATG7 and P62 silencing

Silencing cell line was performed using the CRISPR-Cas9 technology using lentiviral vectors. The lentivirus vectors, pLenti CRISPR (pXPR) vector expressing Cas9 and the single guide (sg) RNAs targeting ATG5, ATG7, SQSTM1 or non-specific sgRNA control, were produced as previously reported [24]. For transduction experiments, Hep3B and SW480 cells were seeded in 6 well plates at a density of 100,000 cells/well and incubated for 24 h in DMEM medium supplemented with 10% heat-inactivated fetal bovine serum (FBS) (Invitrogen) and 50 μg/ml of vancomycin (Sigma Aldrich France) at 37°C in a 5% $CO_2$ humidified atmosphere. The culture medium was then removed and volumes corresponding to a multiplicity of infection (MOI) of 1 virus/cell in FBS-free renewed medium were added to each cell culture well for 12 h. Then, FBS was added to a final concentration of 10% for another 12 h. The medium was then completely renewed with fresh medium with FCS and incubation was continued for 3 days. The cells were seeded in T25 tissue culture flasks. After adherence, the cells were cultured for 5 days in the presence of puromycin (2 μg/ml) to specifically select cells having integrated the sequences of interest and to establish a stable knock-out cell line. This genome-editing approach leads to a clonal cell population comprising 85% of the knock-out (KO) cells, as verified by immunofluorescence analyses with subsequent counting.

## Bacteria & coculture experiments

*H. hepaticus* strain 3B1/Hh-1 and its corresponding isogenic CDT mutant strains lacking CDT activity were cultivated as previously described [6]. Bacterial suspensions were prepared in brucella broth at the optical density (600 nm) adjusted to 0.6 corresponding to a concentration of $2.8x10^8$ colony forming units (CFU)/ml. Coculture experiments were performed at a MOI of 100 bacteria/cell for 72 h as previously described [7,8].

   *E. coli* strain DH10B hosting the bacterial artificial chromosome (BAC pks⁻) vector (pks⁻ *E. coli*) or the BAC pks island encoding colibactin (pks⁺ *E. coli*) were cultivated as previously described [52]. Bacterial suspensions were prepared in Luria Bertani broth at the optical density (600 nm) adjusted to 1 corresponding to a concentration of $5.8x10^8$ CFU/ml. Coculture experiments were performed at a MOI of 100 bacteria/cell for 6 h. Then the medium was

removed and fresh medium with gentamycin (200 µg/ml) was added and incubation was continued for 66 h.

## Autophagic flux measurement

Autophagic flux was measured in cells expressing the tandem-tagged mCherry-GFP-LC3 protein with subsequent dot counting [18]. mCherry-EGFP-LC3-expressing lentiviral vector was kindly provided by Prof. Maria S. Soengas (CNIO, Molecular Pathology Program, Madrid, Spain). Lentiviral vectors allowing the production of the tandem-tagged mCherry-GFP-LC3 protein were used to infect the cells at a MOI of 1 virus/cell with subsequent puromycin selection (1 µg/mL for HT29 and Hep3B and 2 µg/mL for SW480) for 14 days. Autophagic flux was determined by quantifying total GFP and mCherry fluorescence dots in individual cells using an automatic measurement of dot detection using ImageJ software from the National Institutes of Health [54] (http://rsbweb.nih.gov/ij/).

## Other materials and methods

The other materials and methods used in the present study were previously reported. They include transcriptomic analysis using the Human GE 4x44K v2 Microarray Kit (Agilent Technologies, Les Ulis, France) with subsequent statistics [7], western blot analysis [8], and immunofluorescence with subsequent image analysis (wide field and confocal imaging) and protein quantification [6,8].

The sequence of the *cdt*B of *H. hepaticus* strain 3B1 fused at its 30 end to three repeats of the human influenza hemagglutinin (HA) epitope and that of its corresponding mutated *cdt*B sequence (A→T transversion at nucleotide 794 [7]) are available in the GenBank database under the accession numbers KT590046 and KT590047, respectively [21,53].

## Supporting information

**S1 Table. Raw data of autophagy-related genes and inflammation-associated genes included in the Human Microarray Kit.** The global expression of human genes was quantified in transduced epithelial intestinal HT29 cells using the Human whole genome GE 4x44K v2 Microarray Kit (Agilent Technologies) following ectopic expression of the active CdtB subunit of the CDT of *H. hepaticus versus* the control tdTomato fluorescent protein (TFP) [7]. Four independent transduction experiments were performed. **(Sheet1)** Raw data of autophagy-related genes from the Human Autophagy Database. **(Sheet2)** Raw data of inflammation-associated genes.
(XLSX)

**S2 Table. Autophagy family members analyzed using the Human GE 4x44K v2 Microarray Kit (Agilent Technologies) and presented in Fig 1.** Autophagy-related genes (from the Human Autophagy Database) regulated by the CdtB subunit of the CDT of *H. hepaticus*. Presented in Fig 1. Gene name/Symbol, UniqueID, Chromosome coordinates, Accession, Probe name (Agilent) and probe number (when different probes were used) and probe sequence.
(XLSX)

**S3 Table. Autophagy and apoptosis family members (S1 Fig) analyzed using the Human GE 4x44K v2 Microarray Kit (Agilent Technologies).** Autophagy- and apoptosis-related genes (from the Human Autophagy Database) regulated by the CdtB subunit of the CDT of *H. hepaticus*. Presented in S1 Fig. Gene name/Symbol, UniqueID, Chromosome coordinates, Accession, Probe name (Agilent) and probe number (when different probes were used) and

probe sequence.
(XLSX)

**S1 Fig. Effects of the CdtB subunit of *Helicobacter hepaticus* on autophagy-associated apo-
ptosis-related genes and Caspase 1 protein. A)** Microarray-based identification of differen-
tially expressed autophagy associated apoptosis-related genes in response to *Helicobacter
hepaticus* CdtB in HT29 intestinal epithelial cells. The gene expression, transduction protocol,
relative gene expression, results presentation and gene selection are described in the legend of
Fig 1. Asterisks denote significant results. P1 and P2 represent the 2 probe names (S3 Table)
used for mRNA quantification. The data presented for BIRC5, CASP3, CXCR4, FAS (probe 1),
MYC, PTEN and TP53 (probe 1) are the results of 40 replicates as 10 probes for each mRNA
were included on the Microarray Kit. Details are presented in S3 Table (name and sequence of
the probes, the corresponding gene name, the genbank accession number, the locus and the
transcript variant). **B)** Images of 3 μm-tissue sections of HT-29- and Hep3B- derived mice
engrafted tumors stained with fluorescent primary antibody to detect Caspase 1 (green) and
DAPI to counterstain the nuclei (blue). Caspase 1 was quantified on a minimum of 200 cells
using the "Integrated Density" measure function of ImageJ. Scale bar, 30 μm. ***p< 0.0001 *ver-
sus* H265L. Abbreviations: AU, arbitrary unit; BID, BH3 Interacting Domain death agonist;
BIRC5, Baculoviral IAP Repeat-Containing 5; CASP, Caspase apoptosis-related cysteine pepti-
dase; CTSD, Cathepsin D; CTSL1, Cathepsin L1; CXCR4, Chemokine (C-X-C motif) Receptor
4; DAPI, 4′, 6′-diamidino-2-phenylindol; FADD, Fas (TNFRSF6)-associated via death domain;
FAS, Fas (TNF receptor superfamily, member 6); IKBKB, Inhibitor of Kappa light polypeptide
gene enhancer in B-cells, Kinase Beta; MYC, v-myc myelocytomatosis viral oncogene homo-
log; PEA15, Phosphoprotein Enriched in Astrocytes 15; PTEN, Phosphatase and Tensin
homolog; TP53, Tumor Protein p53.
(PDF)

**S2 Fig. Images of the CdtB effects on LC3 expression. (A)** Images of colon HT29 and liver
Hep3B following a 72 h doxycycline-induction to induce the expression of the RFP, CdtB of *H.
hepaticus* strain 3B1 and its corresponding mutated CdtB (H265L)[21]. Cells were processed
as in Fig 2B1 and 2C1. **(B)** Images of CdtB- and H265L-expressing colon HT29 and liver
Hep3B following a 72 h doxycycline-induction and treatment with bafilomycin A1 or chloro-
quine. Cells were processed as in Fig 2B2 and 2C2. **(C)** Images of CdtB- and H265L-expressing
colon HT29 and liver Hep3B expressing the tandem-tagged mCherry-GFP-LC3 protein fol-
lowing a 72 h doxycycline-induction. Cells were processed as in Fig 2B3 and 2C3. **(D)** Autop-
hagic flux was measured following a 72 h doxycycline-induction in CdtB- and H265L-colon
SW480 expressing the tandem-tagged mCherry-GFP-LC3 protein with subsequent yellow
(mCherry$^+$/GFP$^+$) and red (mCherry$^+$/GFP$^-$) dot/puncta counting (yellow dots) [18]. The
results are presented as the mean in one representative experiment (performed in triplicate)
out of three. Cells were processed as in Fig 2B3 and 2C3. Images of colon SW480 expressing
the tandem-tagged mCherry-GFP-LC3 protein. ***p< 0.001 *versus* H265L. Scale bar, 20 μm.
Abbreviations: DAPI, 4′, 6′-diamidino-2-phenylindol; CdtB, CdtB of *H. hepaticus* strain 3B1;
H265L, *H. hepaticus* CdtB with the mutation His→Leu at residue 265 involved in catalytic
activity; GFP, green fluorescent protein; P62, P62/SQSTM1.
(PDF)

**S3 Fig. Effects of bacterial genotoxin on P62/SQSTM1 and UNR/CSDE1 localization. A)**
Images of SK-Hep-1 cells following a 72 h coculture with colibactin-producing *Escherichia
coli*. SK-Hep-1 cells were non-infected or infected for 3 days with colibactin-secreting extra-
intestinal pathogenic *E. coli* and its corresponding isogenic mutant. Then, cells were processed

for fluorescent staining with primary antibodies generated against P62/SQSTM1 (red) and UNR/CSDE1 (green) associated with fluorescent labeled-secondary antibodies and DAPI to counterstain the nuclei (blue). SK-Hep-1 cells non infected and infected with *E. coli* that don't secretes colibactin did not show increase P62/SQSTM1 bodies. Thus, only SK-Hep-1 cells infected with colibactin-secreting *E. coli* are shown. **(A1)** SK-Hep-1 cells with distended nucleus without NR formation. **(A2)** and **(A3)** SK-Hep-1 cells with distended nucleus with NR formation. Arrows indicate nucleoplasmic reticulum (NR). Scale bar, 30 μm. As previously demonstrated, NR formation is primarily observed in response to bacterial genotoxin, CDT and colibactin [6]. Thus, images of non-infected cells and cells infected with colibactine-defective mutant strain are not presented. **(B)** *In vivo* detection of nuclear remodeling following a 14 months infection with *H. hepaticus*. Non-transgenic mice were infected with *H. hepaticus* wild type strain 3B1for 14 months [23]. Mice livers were processed and stained as previously reported [6,23]. Image of tissue sections of infected liver stained with primary and fluorescent secondary antibodies: UNR (red), P62/SQSTM1 (green) and DAPI to counterstain the nuclei (blue). The zone without UNR/P62/DAPI staining in the core of NR probably corresponds to a mitochondrion, as previously reported [6]. P62/SQSTM1-labeling generated a significant background noise in the mice liver tumor infiltrates, preventing efficient quantification. As previously demonstrated, NR formation is primarily observed in response to CDT intoxication, *via* its active CdtB subunit [6]. Thus, images of non-infected mice are not presented below. Arrows indicate nucleoplasmic reticulum (NR) enclosing UNR and P62/SQSTM1. Scale bar, 20 μm. Abbreviations: DAPI, 4′, 6′-diamidino-2-phenylindol; P62, P62/SQSTM1. (PDF)

**S4 Fig. Effects of colibactin on interleukin 8 secretion and nuclear factor κB translocation in human hepatic epithelial cells.** Mock-KO, ATG5-KO and ATG7-KO Hep3B cells were infected for 3 days with colibactin-secreting extra-intestinal pathogenic *E. coli* (pks+) and its corresponding isogenic mutant (pks-). **(A)** IL8 secretion was then quantified on cell culture supernatants (Human IL-8/CXCL8 Quantikine ELISA Kit, R&D Systems, Minneapolis, MN, USA) and **(B)** cells were processed for fluorescent staining with primary antibodies generated against the p65 subunit of nuclear factor κB (NF-κB) and DAPI to counterstain the nuclei (blue). Nuclear translocation of the NF-κB was quantified on a minimum of 500 cells using the "Integrated Density" measure function of ImageJ. $^*p < 0.005$, $^{**}p < 0.001$, $^{***}p < 0.0001$. Abbreviations: AU, arbitrary unit; KO, knock-out; ns, non-significant; pks-, bacterial artificial chromosome vector; pks+, bacterial artificial chromosome vector with pks island encoding colibactin. (PDF)

**S5 Fig. Effects of P62/SQSTM1 and MAFB silencing on nuclear remodeling and P62/SQSTM1 bodies following *Helicobacter hepaticus* infection.** Mock-KO, P62/SQSTM1-KO and MAFB-KO Hep3B cells were not infected or infected for 72 h with *H. hepaticus* and its corresponding ΔCDT mutant strain. These cells were processed for fluorescent staining with DAPI to detect the nucleus (blue) and primary anti-P62/SQSTM1 (red) and anti-UNR/CSDE1 (green). Fluorescent staining was observed using wide field fluorescence imaging. The percentage of cells presenting UNR-NR was determined by manually counting the number of nuclei displaying UNR spots in the nucleoplasm. P62/SQSTM1 bodies were quantified using the "Find Maxima" function of ImageJ. The results are presented as the mean in one representative experiment (performed in triplicate) out of three. A minimum of 500 nuclei were analyzed. As previously demonstrated, NR formation is primarily observed in response to CDT intoxication, *via* its active CdtB subunit [6]. Thus, images of non-infected cells are not presented below. **(A)** Effects of P62/SQSTM1 silencing. Images of cells infected with *H. hepaticus*: P62/

SQSTM1 (red), UNR (green) and DAPI (blue). Scale bar, 20 μm. Quantification of UNR-NR-positive cells in Mock-KO and P62/SQSTM1-KO cells infected with *H. hepaticus*. Western blot analysis of the protein expression level of P62/SQSTM1, UNR/CSDE1 and tubulin in Mock-KO and P62/SQSTM1-KO Hep3B non infected cells. **(B)** Effects of MAFB silencing. MAFB silencing was performed as previously reported and lead to a non-clonal cell population comprising 80% of the knock-out (KO) cells for the MAFB gene [33]. Images of in Mock-KO and MAFB-KO cells infected with *H. hepaticus*: P62/SQSTM1 (red), UNR (green), DAPI (blue). Scale bar, 20 μm. *p<0.05, **p< 0.01, ***p< 0.001. Abbreviations: DAPI, 4′, 6′-diamidino-2-phenylindol; ΔCDT, CDT isogenic mutant of *H. hepaticus* strain 3B1; *H.h*, *Helicobacter hepaticus;* KO, knock-out, NI, non-infected; NR, nucleoplasmic reticulum; ns, non-significant, P62, P62/SQSTM1; WT, *H. hepaticus* strain 3B1 = wild type strain.
(PDF)

**S6 Fig. Effects of the CdtB subunit of *Helicobacter hepaticus* on DNA damage and repair.** Images of 3-μm-tissue sections of Hep3B-derived mice engrafted tumors sequentially stained for 53BP1 (green) during a first labeling round and γH2AX (red) during a second labeling round, along with a counterstaining with DAPI (blue). Images 53BP-1/DAPI were captured using the microscope (wide field) that records the coordinates of each image, which allows repositioning on the same area after the γH2AX/DAPI immunodetection. Magnifications of selected areas are shown in boxes. Arrows indicates the overlapping of 53BP1 and γH2AX foci. The slight difference between the fields showing the 53BP1 and γH2AX labeling is due to the sequential repositioning between the sequential labeling. Abbreviations: CdtB, CdtB of *H. hepaticus* strain 3B1; DAPI, 4′, 6′-diamidino-2-phenylindol; H265L, *H. hepaticus* CdtB with the mutation His→Leu at residue 265 involved in catalytic activity.
(PDF)

## Acknowledgments

1) We are grateful to Chloé Alix for technical assistance. We wish to thank James G. Fox (Division of Comparative Medicine and Biological Engineering, Massachusetts Institute of Technology, Cambridge, MA, USA) for supplying the *H. hepaticus* strains. We thanks Prof. Maria S. Soengas (CNIO, Molecular Pathology Program, Madrid, Spain) for kindly providing us the mCherry-EGFP-LC3-expressing lentiviral vector. We thanks Dr. Valérie Prouzet-Mauléon (Univ. Bordeaux, CRISP'edit platform, TBMcore CNRS UMS3427 / INSERM US005, Bordeaux, France) for kindly providing us the P62/SQSTM1silencing lentiviral vector. We are indebted to Jean-Philippe Nougayrède, Eric Oswald and Frédéric Taieb (Institut de Recherche en Santé Digestive, INRA, Toulouse, France) for supplying the *E. coli*-producing colibactin. We are indebted to Harald Wodrich (MFP CNRS UMR 5234, Microbiologie Fondamentale et Pathogénicité, Université de Bordeaux, France) for supplying anti-lamina (used in Figs 5 and 8).

2) Part of this work was presented as oral presentations at the 'XXXI[th] International Workshop on Helicobacter & Microbiota in Inflammation & Cancer' (European Helicobacter Study Group) (Kaunas, Lithuania, 2018) and annual meeting of the 'GEFH—Groupe d'Etudes Français des Helicobacter' (Paris, France).

## Author Contributions

**Conceptualization:** Mojgan Djavaheri-Mergny.

**Data curation:** Wencan He, Lamia Azzi-Martin, Armelle Ménard.

**Formal analysis:** Wencan He, Lamia Azzi-Martin, Valérie Velasco, Armelle Ménard.

**Funding acquisition:** Wencan He.

**Investigation:** Armelle Ménard.

**Methodology:** Lamia Azzi-Martin, Mojgan Djavaheri-Mergny, Armelle Ménard.

**Project administration:** Armelle Ménard.

**Resources:** Pierre Dubus, Armelle Ménard.

**Supervision:** Armelle Ménard.

**Validation:** Wencan He, Lamia Azzi-Martin, Armelle Ménard.

**Visualization:** Lamia Azzi-Martin, Armelle Ménard.

**Writing – original draft:** Armelle Ménard.

**Writing – review & editing:** Philippe Lehours, Pierre Dubus, Mojgan Djavaheri-Mergny, Armelle Ménard.

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
