## [Decision Letter · Decision Letter 0]

30 Sep 2020

Dear Dr. Menard,

Thank you very much for submitting your manuscript "CDT induces pro-survival autophagy and nucleoplasmic reticulum formation concentrating the RNA binding proteins UNR/CSDE1 and P62/SQSTM1" for consideration at PLOS Pathogens. As with all papers reviewed by the journal, your manuscript was reviewed by members of the editorial board and by several independent reviewers. All reviewers felt that your paper is of potential interest, but raised several points and requested additional experiments that need to be addressed. In light of these reviews (below this email), we would like to invite the resubmission of a significantly-revised version that takes into account the reviewers' comments.

We cannot make any decision about publication until we have seen the revised manuscript and your response to the reviewers' comments. Your revised manuscript is also likely to be sent to reviewers for further evaluation.

Sincerely,

Steffen Backert

Guest Editor

PLOS Pathogens

Guy Tran Van Nhieu

Section Editor

PLOS Pathogens

Kasturi Haldar

Editor-in-Chief

PLOS Pathogens

orcid.org/0000-0001-5065-158X

Michael Malim

Editor-in-Chief

PLOS Pathogens

orcid.org/0000-0002-7699-2064

Reviewer's Responses to Questions

**Part I - Summary**

Reviewer #1: The manuscript entitled ”CDT induces pro-survival autophagy and nucleoplasmic reticulum formation concentrating the RNA binding proteins UNR/CSDE1 and P62/SQSTM1” by He et al describes the activation of autophagy in: i) cells exposed to the H. hepaticus expressing a functional cytolethal distending toxin (CDT) or upon ectopic expression of the active CdtB subunit of the toxin; ii) in in vivo xenograft mouse models. This was assessed by several independent assays ranging from transcriptomic analysis (Figure 1) to immunofluorescence-based methods identifying accumulation of LC3 and p62 punctae in intoxicated/infected cells in vitro or in the xenograft models (Figures 2 and 3).

The authors have identified several nuclear/perinuclear-associated structures which accumulate p62 foci:

i) nucleoplasmic reticulum (NR) invagination, characterized by co-localization of p62 and the UNR protein, devoid of DNA (Figures 3C and 4)

ii) Micronuclei-like structure, positive for LC3 and gH2AX, a surrogate marker of DNA damage (Figures 5A to 5D)

iii) p62 aggregates tightly connected to the lamina of the nuclear membrane (Figure 5E).

Chemical inhibition of autophagy with bafilomycin A/chloroquine and knock out (KO) of ATG5 or ATG7 (key mediators of autophagy) strongly reduces the formation of the UNR-positive structures, reduces gH2AX phosphorylation, increases cell death and formation of p62 punctae (Figure 6).

These effects are not exclusive for CDT intoxication, but they are also induced in cells exposed to Escherichia coli expressing another type of bacterial genotoxin (colibactin, encoded by the pks island, Figure 7), indicating that this is a common response to bacterial genotoxins.

The results reported in this manuscript are interesting and support previous published data demonstrating that CDT induces autophagy and promotes cell survival of the intoxicated cells. In this manuscript the authors associate autophagy activation and cell survival with formation of the p62 positive bodies. The are several issues that need to be clarified. The most relevant is the role of the different p62 positive substructures identified in CDT exposed cells and their role in cell survival, as assessed in more detail in point 1 in Major issues, which will strengthen the biological relevance of the findings.

Reviewer #2: Bacterial genotoxins of the gut microbiota or certain pathogens, including cytolethal distending toxin (CDT), may impact human health. In the current report, whole genome microarrays were performed using HT29 intestinal cells transduced with the active CdtB subunit of Helicobacter hepaticus. The authors observed the CdtB-dependent upregulation of mRNAs involved in positive regulation of autophagy concomitant with the downregulation of mRNAs involved in negative regulation of autophagy. A couple of complimentary studies follow using independent approaches such as immunofluorescence of cell lines and xenograft samples, as well as infection studies with cdtB gene mutants. Together, this is an interesting paper, it’s well written and potentially attractive for a broad readership. I have a few comments that should be considered for improvement.

Reviewer #3: In this work, He et al show in cell culture models of infection with two intestinal pathogens (Helicobacter hepaticus and E coli expressing the pks island and associated genotoxin) that the genotoxic activity is associated with induction of autophagy and the formation of nuclear structures the author refer to as nuclear reticulum, which contains not only autophagy receptors but also RNA-binding proteins. They show using knock out cell lines lacking key autophagy genes that autophagy is an adaptation protecting the cells against apoptosis upon genotoxic damage.

The experiments are performed with care and the results are clear. This reviewer is a bit mystified by the findings. Is autophagy a general consequence of DNA damage, also with non-microbial stimuli? Could there be additional (non-genome-associated) consequences of CDT or colibactin intoxication that trigger autophagy? The control experiments described below (section 2) will potentially shed light on this problem. Also, the data are not entirely novel, as a previous publication (also in PLOS Pathogens, 2019, same group) has already shown the formation of nuclear reticulum upon CDT and colibactin intoxication.

**Part II – Major Issues: Key Experiments Required for Acceptance**

Reviewer #1: 1. It is not clear from the text whether the structures identified in CDT exposed cells and presented in Figures 3 to 5 are structurally different, and they have in common only the association with p62. Which one is associated with autophagy induction? This is also quite difficult to assess presently, since the authors only report the reduction of UNR positive cells upon inhibition of autophagy (Figure 6). Are these structures directly involved in activation of the survival signals? I understand that proving causality is a very difficult question to assess in this context. However if one of the function of autophagy, induced by CDT, is to remove the micronuclei-like structure and reduce the inflammatory response as proposed by the authors in the Discussion section, then it would important to assess whether there is an increase of the gH2AX/p62 bodies in the ATG5/7 KO cells, and if this correlates with a higher inflammatory response (e.g. higher production of pro-inflammatory cytokine) and with higher levels of activation of caspase 1 (by the inflammasome) leading to pyroptosis (see also point 3). If this can be proven, the activation of autophagy would have two important implications: promote cell survival and reduce the inflammatory response, which would help the CDT-producing bacterium to create a suitable microenvironment to support its survival and replication.

On the same line, it would also be important to evaluate the changes in the UNR/p62 positive and DAPI negative bodies upon inhibition of autophagy, as done for the % of UNR positive cells in Figure 6A.

2. Figures 6G and 6H: p62 levels and p62 punctae in control and ATG5/7 KO cells. The authors states that “P62/SQSTM1, a strong increase in the protein was observed in ATG5- and ATG7-KO infected cells in response to H. hepaticus infection, compared to the Mock-KO cells infected with H. hepaticus” (Results section, page 7, lines 40-41). From the figure shown in Figure 6H, the difference in the levels of p62 expression among the different cells is not clear. In addition, the number of p62 nuclear foci in the ATG 7 cells is similar to the control cells, and much lower than that observed in the ATG5 KO. This does not reflect the data presented in the quantification graph (Figure 6G). The authors should clarify this issue.

3. Discussion section. The author state that “H. hepaticus CdtB also regulated some genes involved in the ‘apoptosis and autophagy’ pathway and the increase in the mRNA level of apoptosis-regulator proteins and inflammatory caspases suggests CdtB-induced pyroptosis during H. hepaticus infection” (Page 11, lines 23-25). Conversely, the data in presented in this manuscript support a role of autophagy in survival of the CDT exposed cells, therefore I was wondering how these two opposite outcomes can be reconciled.

Reviewer #2: 1.) The M&M section should contain a separate paragraph describing in detail the microarrays, which is missing. I also think the overall microarray data should be available to the readers, and not only a subset (shown in Fig. 1). Please deposit these data in the public domain, for example at the GEO database. They give you an accession number that can be provided to the reviewers and later to the readers. You can embargo this information for the public domain until the paper is published.

2.) The biological significance of the findings in the paper are not clear to me. It seems not really obvious what is the link between the nuclear bodies positively stained for UNR-p62 or p62/gH2AX related to autophagy-associated cell survival caused by CdtB? Eventually, this could reduce the overall inflammatory reaction (seen by the microarrays or cytokine ELISA?) and could block cell death, which helps the infection strategy of the intruding pathogen? This question could be tackled by some experiments, which in turn can provide biological importance to the studies.

Reviewer #3: Major points:

1. Please use one or two positive controls of DNA damage and genomic instability as controls throughout. Topoisomerase inhibitors or etoposide, or hydroxyurea, or UV irradiation might all work. It is important to see whether any kind of DNA damage leading to DNA DSBs will trigger autophagy and nuclear reticulum formation, or whether this is a specific adaptation to bacterial genotoxins. The best control would be another alkylating agent, which would mimic the effects of colibactin.

2. the authors use micronuclei formation as a readout of genomic instability. The most important question is however not addressed with this assay. How does the inhibition of autophagy (in the ATG5 and ATG7 ko) affect micronuclei formation? In this same context, did the authors expect to see reduced yH2AX foci in cells incapable of autophagy? If DNA DSBs are upstream of autophagy induction, does this finding make sense? Not really to this reviewer. A proper quantification of micronuclei (as a fraction of all binucleated cells) is required.

3. gH2AX as the only readout of DNA DSBs is questionable. Please include a second marker (53BP1 or other) to ascertain that the gH2AX signal is indeed indicative of DNA damage. It doesn't look "focal" enough to this reviewer.

**Part III – Minor Issues: Editorial and Data Presentation Modifications**

Reviewer #1: 1. In Figure3C, it is not really possible to judge whether these structures are external to the nuclear compartment and devoid of DNA, since the DAPI staining is not shown.

2. Figure 5, it would be good to have a higher magnification of the gH2AX/p62 bodies.

3. Figure 5 E is not commented in the text, and also for this figure a higher magnification of the structures indicated would help the reader.

Reviewer #2: 1.) The name of the bacterium “Helicobacter hepaticus” should be in the title of the paper.

2.) Figures 2 and S4: Please add size markers to all Western blots.

Reviewer #3: please make sure all axes are properly labeled for easy comprehension of figure content without the need to read the entire legend.

PLOS authors have the option to publish the peer review history of their article (what does this mean?). If published, this will include your full peer review and any attached files.

Reviewer #1: No

Reviewer #2: No

Reviewer #3: No
---

## [Decision Letter · Decision Letter 1]

31 Dec 2020

Dear Dr. Menard,

Thank you very much for submitting your manuscript "The CDT of Helicobacter hepaticus induces pro-survival autophagy and nucleoplasmic reticulum formation concentrating the RNA binding proteins UNR/CSDE1 and P62/SQSTM1" for consideration at PLOS Pathogens. As with all papers reviewed by the journal, your manuscript was reviewed by members of the editorial board and by several independent reviewers. The reviewers appreciated the attention to an important topic. Based on the reviews, we are likely to accept this manuscript for publication, providing that you modify the manuscript according to the review recommendations.

Sincerely,

Steffen Backert

Guest Editor

PLOS Pathogens

Guy Tran Van Nhieu

Section Editor

PLOS Pathogens

Kasturi Haldar

Editor-in-Chief

PLOS Pathogens

orcid.org/0000-0001-5065-158X

Michael Malim

Editor-in-Chief

PLOS Pathogens

orcid.org/0000-0002-7699-2064

Reviewer Comments (if any, and for reference):

Reviewer's Responses to Questions

**Part I - Summary**

Reviewer #1: The authors have satisfactorily replied to most of the questions risen during the first revision. Importantly they have shown preliminary data demonstrating that inhibition of autophagy can be associated with increased production of inflammatory mediators, supporting an interesting line of investigation concerning the link between bacterial genotoxins and reduction of the host inflammatory response.

Reviewer #2: I appreciate very much the revised paper. The authors have satisfactorily commented to my points.

**Part II – Major Issues: Key Experiments Required for Acceptance**

Reviewer #1: (No Response)

Reviewer #2: (No Response)

**Part III – Minor Issues: Editorial and Data Presentation Modifications**

Reviewer #1: There is still one issue which is not clear to me: it is very difficult to assess from the representative micrograph presented in Figure 6H whether there is a decreased of gH2AX in infected cells upon ATG5 or ATG7 knock out (KO). If this is the case, then this would suggest that inhibition of autophagy is associated with alteration of the DNA damage response. It would be good if the authors:

1) show a lower magnification of the representative gH2AX staining micrographs so the reader can appreciate the reduction in the amount of H2AX phosphorylation in the KO cells in Figure 6H.

2) add in the discussion a sentence concerning the possible mechanisms that promote decreased H2AX phosphorylation upon inhibition of autophagy.

Reviewer #2: (No Response)

PLOS authors have the option to publish the peer review history of their article (what does this mean?). If published, this will include your full peer review and any attached files.

Reviewer #1: No

Reviewer #2: No
---

## [Editor Report · Decision Letter 2]

18 Jan 2021

Dear Dr. Menard,

We are pleased to inform you that your manuscript 'The CDT of Helicobacter hepaticus induces pro-survival autophagy and nucleoplasmic reticulum formation concentrating the RNA binding proteins UNR/CSDE1 and P62/SQSTM1' has been provisionally accepted for publication in PLOS Pathogens.

Best regards,

Steffen Backert

Guest Editor

PLOS Pathogens

Guy Tran Van Nhieu

Section Editor

PLOS Pathogens

Kasturi Haldar

Editor-in-Chief

PLOS Pathogens

orcid.org/0000-0001-5065-158X

Michael Malim

Editor-in-Chief

PLOS Pathogens

orcid.org/0000-0002-7699-2064
---

## [Editor Report · Acceptance letter]

26 Feb 2021

Dear Dr Ménard,

We are delighted to inform you that your manuscript, "The CDT of Helicobacter hepaticus induces pro-survival autophagy and nucleoplasmic reticulum formation concentrating the RNA binding proteins UNR/CSDE1 and P62/SQSTM1," has been formally accepted for publication in PLOS Pathogens.

Best regards,

Kasturi Haldar

Editor-in-Chief

PLOS Pathogens

orcid.org/0000-0001-5065-158X

Michael Malim

Editor-in-Chief

PLOS Pathogens

orcid.org/0000-0002-7699-2064